# DiffSplat: Repurposing Image Diffusion Models for Scalable Gaussian Splat Generation

**Chenguo Lin[1], Panwang Pan[†2], Bangbang Yang[2], Zeming Li[2], Yadong Mu[‡1]**
[1]Peking University, [2]ByteDance
**https://chenguolin.github.io/projects/DiffSplat**

## Abstract

Recent advancements in 3D content generation from text or a single image struggle with limited high-quality 3D datasets and inconsistency from 2D multi-view generation. We introduce DiffSplat, a novel 3D generative framework that natively generates 3D Gaussian splats by taming large-scale text-to-image diffusion models. It differs from previous 3D generative models by effectively utilizing web-scale 2D priors while maintaining 3D consistency in a unified model. To bootstrap the training, a lightweight reconstruction model is proposed to instantly produce multi-view Gaussian splat grids for scalable dataset curation. In conjunction with the regular diffusion loss on these grids, a 3D rendering loss is introduced to facilitate 3D coherence across arbitrary views. The compatibility with image diffusion models enables seamless adaptions of numerous techniques for image generation to the 3D realm. Extensive experiments reveal the superiority of DiffSplat in text- and image-conditioned generation tasks and downstream applications. Thorough ablation studies validate the efficacy of each critical design choice and provide insights into the underlying mechanism.

## 1 Introduction

Generating 3D content from a single image or text is a long-standing challenge with a wide range of applications, such as game design, digital arts, human avatars, and virtual reality. It is a highly ill-posed problem that requires reasoning the unseen parts of any object in the 3D space only from a single view or textual descriptions, posing a great challenge to both fidelity and generalizability.

With the development of diffusion generative models (Sohl-Dickstein et al., 2015; Ho et al., 2020), recent works train conditional 3D generative networks directly on datasets of various 3D representations (Nichol et al., 2022; Jun & Nichol, 2023; Cao et al., 2024; He et al., 2024; Zhang et al., 2024b), as demonstrated in Figure 1 (1), or only using 2D supervision with the help of differentiable rendering techniques (Anciukevičius et al., 2023; Karnewar et al., 2023b; Szymanowicz et al., 2023; Xu et al., 2024d) as in Figure 1 (2). Despite 3D consistency, they are limited by the supervision scale and can't utilize 2D priors from abundant pre-trained models. Current advanced generalizable 3D content creation methods (Li et al., 2024a; Wang et al., 2024; Tang et al., 2024) reconstruct implicit 3D representations from generated multi-view images using pretrained 2D diffusion models (Wang & Shi, 2023; Shi et al., 2023; Voleti et al., 2024), as illustrated in Figure 1 (3). Although these two-stage methods can reconstruct high-quality 3D content from multi-view posed images, the synthesis pipeline collapses and fails to produce faithful results when generated images from upstreamed 2D multi-view diffusion models are of poor quality or inconsistency.

To overcome the drawbacks of previous works, we present DiffSplat, a novel 3D generative framework that exhibits multi-view consistency and effectively leverages generative priors from large-scale image datasets. We adopt 3D Gaussian Splatting (3DGS) (Kerbl et al., 2023) as the 3D content representation for its efficient rendering and quality balance. Instead of relying on time-consuming per-instance optimization to obtain 3D datasets for training (He et al., 2024; Zhang et al., 2024b), we represent a 3D object by a set of well-structured splat 2D grids. During the training stage, these grids can be instantly regressed from multi-view images in less than 0.1 seconds, facilitating scalable and high-quality 3D dataset curation. Each Gaussian splat in 2D grids holds properties that imply object texture and structure. Noting that image diffusion models trained on web-scale datasets are capable

---

†: Project lead; ‡: Corresponding author.

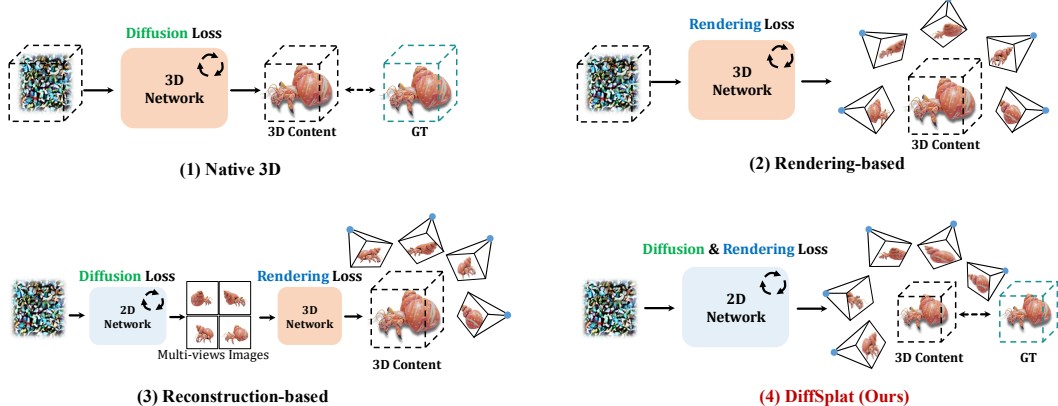

Figure 1: **Comparison with Previous 3D Diffusion Generative Models**. (1) Native 3D methods and (2) rendering-based methods encounter challenges in training 3D diffusion models from scratch with limited 3D data. (3) Reconstruction-based methods struggle with inconsistencies in generated multi-view images. In contrast, (4) DIFFSPLAT leverages pretrained image diffusion models for the direct 3DGS generation, effectively utilizing 2D diffusion priors and maintaining 3D consistency. "GT" refers to ground-truth samples in a 3D representation used for diffusion loss computation.

of 3D geometry estimation (Li et al., 2024b; Ke et al., 2024; Fu et al., 2024), we find that by treating reconstructed Gaussian splat 2D grids as images in a special style, we can **harness the power of pretrained 2D image diffusion models for direct 3DGS generation**.

Specifically, open-sourced latent diffusion models (Rombach et al., 2022; Podell et al., 2024; Chen et al., 2024c;b; Esser et al., 2024) trained on web-scale image datasets are repurposed to directly generate Gaussian splat properties for 3D content creation. To ensure that reconstructed splat grids have the same shape as the input latents of image diffusion models, we also fine-tune their VAEs to compress Gaussian splat properties into a similar latent space, which are coined as *splat latents*. During the training of DIFFSPLAT, as shown in Figure 1 (4), in addition to a standard diffusion loss on splat latents, which resembles regular image diffusion models, we propose to incorporate a rendering loss to enable the generative model to function in 3D space and facilitate 3D consistency, since Gaussian splat properties are processed in the network and can be differentially rendered in arbitrary views. Furthermore, thanks to the minimal modifications on 2D denoising network architectures, various pretrained text-to-image diffusion models can serve as the base model for DIFFSPLAT, and techniques proposed for them can be seamlessly adapted to the realm of 3D generation within our framework, establishing a bridge between 3D content creation and the image generation community.

Our contributions can be summarized as follows:

- A novel 3D generative framework that directly generates 3D Gaussian splats by fine-tuning image diffusion models, effectively utilizing 2D priors and maintaining 3D consistency.
- Abundant pretrained image diffusion models and associated techniques can be adapted for the proposed model, seamlessly connecting 3D generation and the image community.
- Extensive experiments demonstrate the superior performance of the proposed method, and ablation studies are conducted to analyze the effectiveness of each design choice.

## 2  RELATED WORK

**Native 3D Generative Models**  Diffusion-based models (Ho et al., 2020; Liu et al., 2023) have emerged as the de facto approach for generative models. The most straightforward solution for 3D content creation is to train a 3D denoising network on 3D datasets as shown in Figure 1 (1), termed as "native 3D generative models". Numerous native 3D methods are proposed for explicit 3D representations, such as voxel (Sanghi et al., 2023; Ren et al., 2024) and point cloud (Nichol et al., 2022; Mo et al., 2023), which are readily available but often result in poor visual quality. On the other hand, implicit representations like implicit functions (Jun & Nichol, 2023; Liu et al., 2024a; Zhang et al., 2024d; Li et al., 2024c; Chen et al., 2024d), triplanes (Cao et al., 2024; Liu et al., 2024b; Zhang et al., 2024a; Wu et al., 2024) and 3DGS (Henderson et al., 2024; He et al., 2024; Zhang

et al., 2024b) offer more faithful appearances but are usually inaccessible and require extra time-intensive preprocessing, making it impractical to maintain a large-scale dataset. Moreover, native 3D generative models face limitations in leveraging pretrained 2D models, posing great challenges to the quality and scale of 3D datasets, as well as the efficiency of 3D network training from scratch.

**Rendering-based 3D Generative Models** Instead of relying on time-consuming 3D dataset curation, with differentiable rendering techniques (Mildenhall et al., 2020), some works propose to train 3D generative models with only 2D supervision, which are called "rendering-based generative models" and shown in Figure 1 (2). These methods aim to denoise images rendered from a corrupted implicit 3D representation in the absence of ground-truth clean 3D data. HoloDiffusion (Karnewar et al., 2023b) and HoloFusion (Karnewar et al., 2023a) propose bootstrapped latent diffusion strategy, in which feature volumes are denoised twice to solve the problem that there is no ground-truth rendering volume for radiance fields. Compared to RenderDiffusion (Anciukevičius et al., 2023) that reconstructs clean triplane features from single-view noised images, Viewset Diffusion (Szymanowicz et al., 2023) and GIBR (Anciukevicius et al., 2024) denoise multi-view images simultaneously, ensuring coherent and plausible appearance and geometry for the intermedia rendering volume from sufficient viewpoints. DMV3D (Xu et al., 2024d) further scales up to highly diverse datasets containing nearly 1 million objects. However, it is extremely computationally intensive and costs 128 A100 GPUs for 7 days to train a unified model for 2D denoising and 3D reconstruction without any 2D generative prior, suffering from the same drawback of native 3D methods.

**Reconstruction-based 3D Generative Models** By scaling up network parameters and training datasets, Large Reconstruction Model (LRM) (Hong et al., 2024b; Tochilkin et al., 2024), a feed-forward Transformer encoder (Vaswani et al., 2017), can reconstruct a triplane field (Chan et al., 2022) from a single image. It only needs multi-view images for supervision akin to rendering-based methods, but operates deterministically rather than for denoising. As presented in Figure 1 (3), reconstruction-based methods leverage 2D priors by utilizing frozen image diffusion models to generate multiview images (Shi et al., 2024; Wang & Shi, 2023; Shi et al., 2023; Voleti et al., 2024; Han et al., 2024) with generalizable 3D reconstruction models. Instant3D (Li et al., 2024a) follows LRM and produces triplane features from posed images in $2 \times 2$ grids generated by a fine-tuned large image diffusion model (Podell et al., 2024). Some methods (Wang et al., 2024; Wei et al., 2024; Xu et al., 2024b; Siddiqui et al., 2024; Boss et al., 2024) replace the rendering representation from triplane to FlexiCubes (Shen et al., 2021; 2023) or VolSDF (Yariv et al., 2021) to directly extract meshes. Other methods produce Gaussian attributes from pixels (Tang et al., 2024; Xu et al., 2024c; Zhang et al., 2024c), learnable tokens (Chen et al., 2024a) or latent features (Pan et al., 2024). However, all these methods regard the multi-view diffusion model as an independent plug-and-play module, so 3D generation is conducted as a two-stage proceeding, which requires extra network parameters and may collapse due to the 3D inconsistency in generated images.

## 3 METHOD

Motivated by the effectiveness of web-scale pretrained image diffusion models in estimating 3D geometry attributes, such as depth (Stan et al., 2023; Ke et al., 2024), coordinates (Li et al., 2024b; Xu et al., 2024a) and normal (Long et al., 2024; Fu et al., 2024), our goal in this work is **taming image diffusion models to directly generate 3D content**. As illustrated in Figure 2, the proposed method consists of three parts: (1) scalable 3D data curation by structured splat reconstruction (Sec. 3.1), (2) splat latents (Sec. 3.2), and (3) the generative model DIFFSPLAT (Sec. 3.3).

### 3.1 DATA CURATION: STRUCTURED SPLAT RECONSTRUCTION

Inspired by generalizable 3DGS reconstruction techniques (Szymanowicz et al., 2024; Charatan et al., 2024), we utilize a set of structured multi-view Gaussian splat grids to represent a 3D object. Specifically, given $V_{in}$ posed images in $\mathbb{R}^{3 \times H \times W}$, a small network $F_\theta$ can regress per-pixel splat from these contextualized images in **under 0.1 seconds**, and is trained by the rendering loss $\mathcal{L}_{render}$:

$$\mathcal{L}_{render}(\mathcal{G}) := \frac{1}{V} \sum_{v=1}^{V} \left( \mathcal{L}_{MSE}(I_v, I_v^{GT}) + \lambda_p \cdot \mathcal{L}_{LPIPS}(I_v, I_v^{GT}) + \lambda_\alpha \cdot \mathcal{L}_{MSE}(M_v, M_v^{GT}) \right), \quad (1)$$

where $I_v$ and $M_v$ are respectively RGB images and silhouette masks differentially rendered from predicted 3D Gaussian primitives $\mathcal{G} := \{\mathbf{g}_i\}_{i=1}^{N}$ via efficient rasterization technique (Kerbl et al.,

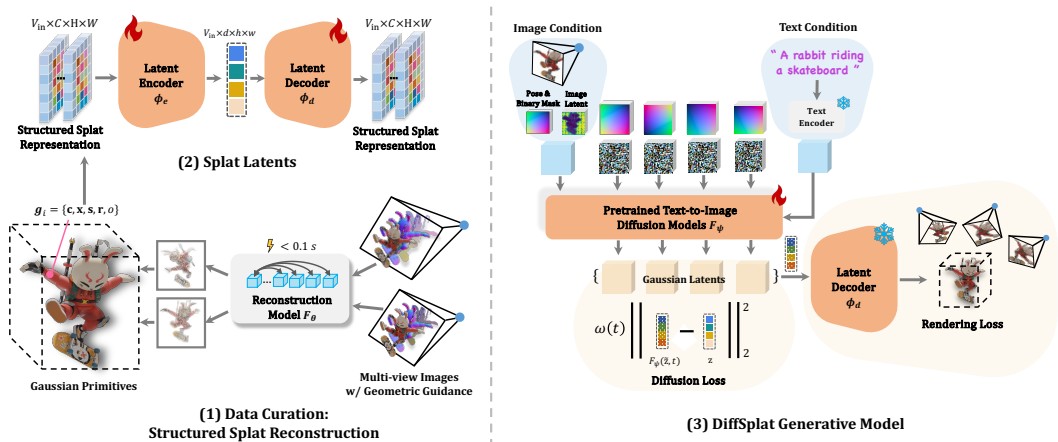

Figure 2: **Method Overview**. (1) A lightweight reconstruction model provides high-quality structured representation for "pseudo" dataset curation. (2) Image VAE is fine-tuned to encode Gaussian splat properties into a shared latent space. (3) DIFFSPLAT is natively capable of generating 3D contents by image and text conditions utilizing 2D priors from text-to-image diffusion models.

2023) across random $V$ viewpoints, and $N = V_{\text{in}} \times H \times W$ due to the pixel-aligned prediction. $\mathcal{L}_{\text{MSE}}$ and $\mathcal{L}_{\text{LPIPS}}$ stands for mean squared error loss and VGG-based perceptual loss (Zhang et al., 2018). "GT" denotes ground-truth data for supervision, and $\lambda_p, \lambda_\alpha \in \mathbb{R}^+$ are hyper-parameters to adjust the relative importance of three different losses.

Each Gaussian primitive $\mathbf{g}_i \in \mathbb{R}^{12}$ is parameterized by its RGB color $\mathbf{c} \in \mathbb{R}^3$, location $\mathbf{x} \in \mathbb{R}^3$, scale $\mathbf{s} \in \mathbb{R}^3$, rotation quaternion $\mathbf{r} \in \mathbb{R}^4$ and opacity $o \in \mathbb{R}$. To simplify the parameterization and regulate the distribution of primitives, location $\mathbf{x}$ is determined by depth $d \in \mathbb{R}$, camera intrinsic $\mathbf{K} \in \mathbb{R}^{3\times3}$ and extrinsic matrices (rotation $\mathbf{R} \in SO(3)$ and translation $\mathbf{t} \in \mathbb{R}^3$), $\mathbf{x} := \mathbf{R}^\top \mathbf{K}^{-1}[\mathbf{u}|d] - \mathbf{t}$. $[\mathbf{u}|d]$ is homogeneous pixel coordinates $\mathbf{u} \in \mathbb{R}^2$ with depth. For numerical values of Gaussian splat properties to be confined within $[0, 1]$ in preparation of diffusion-based generation, outputs of $F_\theta$ are all activated by the sigmoid function $\sigma(\cdot)$, except for $\mathbf{r}$, which is $L_2$-normalized to yield unit quaternions. RGB color $c$ and opacity $o$ are already supposed to be in $[0, 1]$. Raw scale $\hat{\mathbf{s}}$ is linearly interpolated with predefined values $s_{\text{min}}$ and $s_{\text{max}}$ (Xu et al., 2024c):

$$\mathbf{s} := s_{\text{min}} \cdot \sigma(\hat{\mathbf{s}}) + s_{\text{max}} \cdot (1 - \sigma(\hat{\mathbf{s}})). \tag{2}$$

Raw depth $\hat{d}$ is defined to be relative to the image plane, and objects in our datasets can be normalized into a $[-1, 1]^3$ cube, allowing its value to be restricted in $[0, 1]$ by the sigmoid function:

$$d := 2 \cdot \sigma(\hat{d}) - 1 + \|\mathbf{t}\|_2. \tag{3}$$

Different from previous reconstruction-based methods (Tang et al., 2024; Xu et al., 2024c; Zhang et al., 2024c), besides multi-view posed RGB images, we also incorporate corresponding coordinate maps (Shotton et al., 2013) and normal maps as additional inputs for auxiliary geometric guidance. Note that these extra inputs are only employed to enhance the reconstruction quality of Gaussian splat grids and are not required during the generation stage. Ablation study is conducted to assess the efficacy of geometric guidance alongside RGB images in Sec. 4.5.1.

## 3.2 SPLAT LATENTS

Aforementioned designs allow for a high-quality representation of 3D objects using multi-view splat grids $\mathcal{G} := \{\mathbf{G}_i\}_{i=1}^{V_{\text{in}}}$, which are structured for image-like processing, obtained efficiently, and confined within a suitable numerical range. To encode Gaussian splat grids to image diffusion latent space, the most straightforward idea is to split them into multiple feature groups with 3 channels to analog RGB images, and process them separately. However, this results in poor auto-encoding quality, as encoded latents display notable deviations from natural images, as shown in Sec. 4.5.1.

Instead, we duplicate the columns and rows of pretrained input and output convolution weights 4 times respectively to match the feature dimensions of Gaussian splat grids $\mathbf{G}_i \in \mathbb{R}^{12 \times H \times W}$. VAEs for latent image diffusion models (Rombach et al., 2022; Podell et al., 2024; Esser et al., 2024) are

then fine-tuned to auto-encode each Gaussian splat grid independently with both reconstruction loss and rendering loss:

$$\mathcal{L}_{\text{VAE}} := \frac{1}{V_{\text{in}}} \sum_{i=1}^{V_{\text{in}}} \left( \mathcal{L}_{\text{MSE}}(\tilde{\mathbf{G}}_i, \mathbf{G}_i) \right) + \lambda_r \cdot \mathcal{L}_{\text{render}}(\tilde{\mathcal{G}}), \tag{4}$$

where $\tilde{\mathbf{G}}_i := D_{\phi_d}(E_{\phi_e}(\mathbf{G}_i))$ is auto-encoded $\mathbf{G}_i$ by the VAE's encoder $E_{\phi_e}$ and decoder $D_{\phi_d}$, and $\mathcal{L}_{\text{render}}$ is computed across $V$ random views as presented in Equation 1. $\lambda_r$ is the weighting term for the rendering loss. Encoded Gaussian splat grids from $V_{\text{in}}$ input views $\mathbf{z} := \{\mathbf{z}_i\}_{i=1}^{V_{\text{in}}} = \{E_{\phi_e}(\mathbf{G}_i)\}_{i=1}^{V_{\text{in}}}$ are referred to as ***splat latents***, which undergoes diffusion and denoising process in DIFFSPLAT. Rendering loss $\mathcal{L}_{\text{render}}$ is significant for high-quality splat latent auto-encoding, and quantitative evaluations are provided in Sec. 4.5.1.

### 3.3 DIFFSPLAT GENERATIVE MODEL

#### 3.3.1 MODEL ARCHITECTURE

Given a set of multi-view splat latents $\mathbf{z} = \{\mathbf{z}_i\}_{i=1}^{V_{\text{in}}}$ structured as 2D grids, two widely adopted manners for multi-view image generation are explored in this work to simultaneously generate $\mathbf{z}$ as a whole from text prompts or single-view images, coined as ***view-concat*** and ***spatial-concat***.

In the ***view-concat*** manner, $V_{\text{in}}$ splat latents of an objects, shaped as $\mathbb{R}^{d \times h \times w}$, are treated like video frames and concatenated along the view dimension into $\mathbb{R}^{V_{\text{in}} \times d \times h \times w}$ and processed individually by the denoising network, except in the self-attention module, where they are reshaped into $\mathbb{R}^{(V_{\text{in}} \cdot h \cdot w) \times d}$ and treated as an integral sequence (Long et al., 2024; Shi et al., 2024; Kant et al., 2024; Gao et al., 2024b). While for the ***spatial-concat*** manner, splat latents are organized into an $r \times c$ grid along the height and width dimensions, resulting in a shape of $\mathbb{R}^{d \times (r \cdot h) \times (c \cdot w)}$, where $V_{\text{in}} \equiv r \times c$ (Shi et al., 2023; Li et al., 2024a; Gao et al., 2024a). In both manners, Plücker embeddings (Sitzmann et al., 2021) are concatenated along the feature dimension with respective splat latents, enabling a dense encoding of relative camera poses. It facilitates better flexibility in viewpoint selection and reduces requirements for multi-view datasets. The only new parameters introduced to pretrained models are zero-initialized new columns in the input convolution weights for Plücker embeddings. These model designs result in minimal modifications and generalizability across various text-to-image diffusion models (Rombach et al., 2022; Podell et al., 2024; Chen et al., 2024c;b; Esser et al., 2024).

Unlike multi-view image diffusion models (Li et al., 2024a; Kant et al., 2024), it's not feasible for text-conditioned DIFFSPLAT to simply denoise other views except for the input image view for image-conditioned generation, as the input condition (pixels) and generated outputs (splat properties) are in different domains. For ***view-concat***, the original VAE-encoded input image is concatenated along the view dimension, with additional dense binary masks concatenated along the feature dimension to distinguish between image and splat latents. For ***spatial-concat***, the input image is padded with a blank background to form an $r \times c$ grid, and then concatenated along the feature dimension after the image VAE encoding. Ablation study on the multi-view manners for both text- and image-conditioned 3DGS generation tasks is conducted in Sec. 4.5.2.

#### 3.3.2 TRAINING OBJECTIVES

DIFFSPLAT $F_\psi$ can be trained with the regular diffusion loss $\mathcal{L}_{\text{diff}}$, which aims to denoise corrupted splat latents $\tilde{\mathbf{z}} := \texttt{AddNoise}(\mathbf{z}, \boldsymbol{\epsilon}, t)$ from a randomly sampled noise level $t$:

$$\mathcal{L}_{\text{diff}} := \omega(t) \cdot \| F_\psi(\tilde{\mathbf{z}}, t) - \mathbf{z} \|_2^2. \tag{5}$$

Here, $\texttt{AddNoise}(\cdot)$ corrupts the sample $\mathbf{z}$ with random Gaussian noises $\boldsymbol{\epsilon} \sim \mathcal{N}(\mathbf{0}, \mathbf{1})$ at the noise level $t$ (Ho et al., 2020; Nichol & Dhariwal, 2021; Karras et al., 2022; Liu et al., 2023), and $\omega(t)$ is the weighting term of diffusion loss at different noise levels (Song et al., 2021; Salimans & Ho, 2022; Hang et al., 2023; Kingma & Gao, 2023). Text and image conditions are omitted for concision.

However, optimizing solely with $\mathcal{L}_{\text{diff}}$, similar to multi-view image diffusion models (Shi et al., 2024; 2023; Li et al., 2024a), does not guarantee 3D consistency. This limitation stems from the model essentially operating in the 2D space of Gaussian splat grids, and the stereo correspondences among $V_{\text{in}}$ viewpoints are learned implicitly during the fine-tuning of image diffusion models, which

are originally trained on single-view natural images. Therefore, it makes the fine-tuning process less straightforward for achieving consistent learning in a 3D context. Moreover, treating splat latents as ground-truth samples, which are compressed after a lightweight reconstruction model, also limits the upper bound of the generative model, as real multi-view datasets are not involved in the training process, relying completely on the results from reconstruction and auto-encoding models.

Recognizing that splat latents are processed during the diffusion process, not as pixels but as a natural 3D representation that can be efficiently rendered from arbitrary views, we propose to incorporate an additional rendering loss $\mathcal{L}_{\text{render}}$, as defined in Equation 1, alongside $\mathcal{L}_{\text{diff}}$, where denoised splat latents are decoded back into Gaussian splat properties and rendered from random $V$ viewpoints, with supervision from ground-truth multi-view images. The final training objective is:

$$\mathcal{L}_{\text{DIFFSPLAT}} := \lambda_{\text{diff}} \cdot \mathcal{L}_{\text{diff}} + \lambda_{\text{render}} \cdot \omega_r(t) \cdot \mathcal{L}_{\text{render}}(D_{\phi_d}(F_\psi(\tilde{\mathbf{z}}, t))), \tag{6}$$

where $\omega_r(t)$ is the weighting term of the rendering loss at different noise levels, and $\lambda_{\text{diff}}, \lambda_{\text{render}}$ are used to adjust the importance of two types of losses.

Notably, by setting $\lambda_{\text{diff}} = 0$, DIFFSPLAT effectively becomes a rendering-based model, but denoising splat latents instead of pixels. On the other hand, by setting $\lambda_{\text{render}} = 0$, DIFFSPLAT transforms into a "pseudo" native 3D model by treating splat latents as a pseudo ground-truth 3D representation. The effectiveness of the two types of losses is evaluated in Sec. 4.5.2.

## 4 EXPERIMENTS

### 4.1 EXPERIMENTAL SETTINGS

All our models in this work are trained on G-Objaverse (Qiu et al., 2024), a high-quality subset of Objaverse (Deitke et al., 2023) and comprising images from `38` different views of around `265K` 3D objects. Captions of these 3D objects are provided by Cap3D (Luo et al., 2023; 2024). To quantitatively evaluate the performance of text-conditioned generation, `300` text prompts from T3Bench (He et al., 2023), describing a single object, a single object with surroundings and multiple objects, are employed as conditions. CLIP similarity score (Radford et al., 2021) and CLIP R-Precision (Park et al., 2021) based on `ViT-B/32` are used to measure the alignment of input prompts and rendered images, and ImageReward (Xu et al., 2023) is used to reflect human aesthetic preference. For both reconstruction and image-conditioned generation task, `300` objects from the unseen GSO (Downs et al., 2022) dataset are randomly selected and rendered to serve as ground-truth images, which are then compared with rendered images from reconstructed or generated 3D contents in terms of PSNR, SSIM and LPIPS (Zhang et al., 2018) metrics. All metrics are averaged across different viewpoints for 3D-aware evaluation. Implementation details are provided in Appendix A.

### 4.2 TEXT-CONDITIONED GENERATION

**Baselines**  Four state-of-the-art open-sourced methods that support native text-to-3D generation are evaluated, where GVGEN (He et al., 2024) uses Gaussian volume to represent 3D objects, while triplane-based NeRF is used in LN3Diff (Lan et al., 2024), DIRECT-3D (Liu et al., 2024b) and 3DTopia (Hong et al., 2024a). Two reconstruction-based methods LGM (Tang et al., 2024) and GRM (Xu et al., 2024c) can support text-to-3D generation by associating with an open-source text-conditioned multi-view diffusion model (Shi et al., 2024).

**Results and Comparisions**  As demonstrated in Table 1 and Figure 3, DIFFSPLAT exhibits the best prompt alignment and visual quality among cutting-edge text-conditioned 3D generation methods, especially for complex prompts. In contrast, 3D native methods struggle to match text prompts due to limited text-3D pairs for training from scratch, while reconstruction-based methods suffer from multi-view diffusion inconsistency, particularly for objects with surroundings or other objects. More visualization results are provided in Appendix Figure 9, 10 and 11.

### 4.3 IMAGE-CONDITIONED GENERATION

**Baselines**  Two up-to-date native 3D models that support image-conditioned generation are compared here: the concurrent work 3DTopia-XL (Chen et al., 2024d) and LN3Diff (Lan et al., 2024). Six advanced reconstruction-based methods for single image-conditioned generation are also evaluated, including three Gaussian Splatting-based (Kerbl et al., 2023) methods: LGM (Tang et al.,

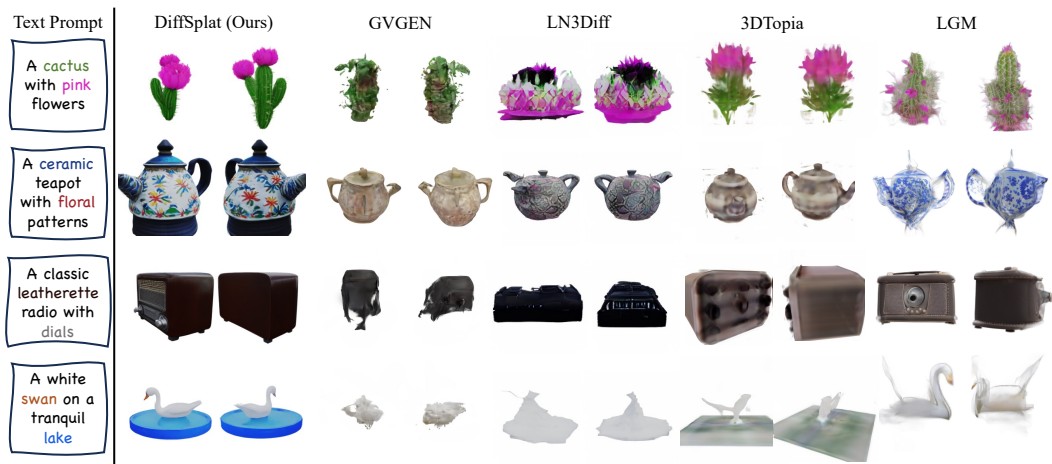

Figure 3: **Qualitative Results and Comparisons on Text-conditioned 3D Generation**. More visualizations of DIFFSPLAT results are provided in Appendix Figure 9, 10 and 11.

Table 1: Quantive evaluations on T3Bench prompts for text-conditioned generation. † indicates reconstruction-based methods that require additional text-conditioned multi-view generative models.

| | | DIFFSPLAT | GVGEN | LN3Diff | DIRECT-3D | 3DTopia | LGM† | GRM† |
|---|---|---|---|---|---|---|---|---|
| Single Object | ↑ CLIP Sim.% | **30.95** | 23.66 | 24.36 | 24.80 | 25.55 | 29.96 | 28.19 |
| | ↑ CLIP R-Prec.% | **81.00** | 23.25 | 27.25 | 30.75 | 34.50 | 78.00 | 64.75 |
| | ↑ ImageReward | **-0.491** | -2.156 | -2.008 | -2.005 | -1.998 | -0.720 | -1.337 |
| Single Object w/ Sur. | ↑ CLIP Sim.% | **30.20** | 22.65 | 22.75 | 23.05 | 24.31 | 27.79 | 26.24 |
| | ↑ CLIP R-Prec.% | **80.75** | 26.75 | 22.00 | 25.75 | 39.00 | 55.00 | 51.25 |
| | ↑ ImageReward | **-0.674** | -2.251 | -2.244 | -2.191 | -2.230 | -1.772 | -1.869 |
| Multiple Objects | ↑ CLIP Sim.% | **29.46** | 21.48 | 21.65 | 21.89 | 22.88 | 27.07 | 24.33 |
| | ↑ CLIP R-Prec.% | **69.50** | 8.00 | 8.75 | 7.75 | 16.50 | 51.00 | 26.50 |
| | ↑ ImageReward | **-0.849** | -2.272 | -2.267 | -2.249 | -2.225 | -1.731 | -2.116 |

2024), GRM (Xu et al., 2024c) and LaRa (Chen et al., 2024a), and two FlexiCube-based (Shen et al., 2023) methods: CRM (Wang et al., 2024) and InstantMesh (Xu et al., 2024b). Image generative models for these reconstruction methods are selected following their original implementations.

**Results and Comparisons** Single image-conditioned generation performance on the GSO dataset is assessed in Table 2, and qualitative results on in-the-wild images are presented in Figure 4 and Appendix Figure 12, 13 and 14. DIFFSPLAT delivers accurate 3D content aligned with input images while maintaining strong geometric fidelity compared to other state-of-the-art methods.

### 4.4 APPLICATION: CONTROLLABLE GENERATION

Thanks to the compatibility of DIFFSPLAT with image diffusion models, numerous techniques initially developed for image generation can be easily adapted for 3D applications. Here, we explore ControlNet (Zhang et al., 2023) to facilitate controllable generation guided by flexible formats alongside text prompts in Figure 5. DIFFSPLAT can generate diverse and high-quality 3D assets that accurately respond to different control inputs, such as normal and depth maps, and Canny edges, while faithfully reflecting the text conditions. Moreover, while most previous reconstruction methods cannot incorporate text understanding, the flexible conditioning design allows DIFFSPLAT to perform text-guided reconstruction from single-view ambiguous images, as shown in Appendix Figure 8.

### 4.5 ABLATION AND ANALYSIS

We carefully investigate each design choice for splat latent reconstruction and DIFFSPLAT 3D generation in this subsection. Ablation studies are conducted based on Stable Diffusion V1.5 (SD1.5) (Rombach et al., 2022) unless otherwise specified.

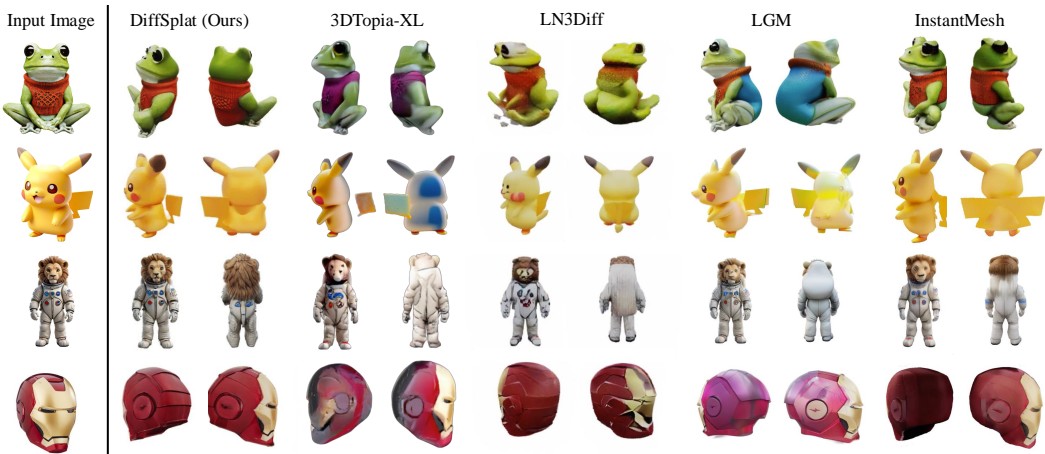

Figure 4: **Qualitative Results and Comparisons on Image-conditioned 3D Generation**. More visualizations of DIFFSPLAT results are provided in Appendix Figure 12, 13 and 14.

Table 2: Quantitative evaluations on GSO for image-conditioned generation. [†] indicates reconstruction methods that require additional image generation models for single image-to-3D generation.

|  | DIFFSPLAT | 3DTopia-XL | LN3Diff | LGM[†] | GRM[†] | LaRa[†] | CRM[†] | InstantMesh[†] |
|---|---|---|---|---|---|---|---|---|
| ↑ PSNR | **22.91** | 17.27 | 16.67 | 18.25 | 19.65 | 18.87 | 18.56 | 19.14 |
| ↑ SSIM | **0.892** | 0.840 | 0.831 | 0.841 | 0.869 | 0.852 | 0.855 | 0.876 |
| ↓ LPIPS | **0.107** | 0.175 | 0.177 | 0.166 | 0.141 | 0.202 | 0.149 | 0.128 |

### 4.5.1 SPLAT LATENT RECONSTRUCTION

**Reconstruction Inputs** Effectiveness of geometric guidance for the reconstruction model is validated in Table 3. Gaussian Splatting-based large reconstruction model LGM (Tang et al., 2024) and GRM (Xu et al., 2024c) are also evaluated with sparse-view RGB images for comparison. Although with much fewer parameters, with the help of coordination and normal maps, the proposed lightweight reconstruction model can instantly provide high-quality Gaussian splat grids as "pseudo" ground-truth representation for 3D generation. Coordination maps explicitly indicate the positions of Gaussian primitives, thus providing more effective geometric guidance than normal maps.

**Auto-encoding Stragegies** We investigate different training strategies for Gaussian splat property auto-encoding in Table 4. Freezing the original image VAE or its encoder results in poor performance, as Gaussian splat properties differ significantly from natural images. Rendering loss plays a crucial role in auto-encoding by ensuring that the VAE is supervised by real datasets rather than being limited by the lightweight reconstruction model, thus enabling the auto-encoded splats to perform slightly better than the reconstructed ones. VAEs from SD1.5 and SDXL (Podell et al., 2024) have a similar performance with the same dimension ($d = 4$) of latent space, while SD3 (Esser et al., 2024) shows improved performance with $d = 16$.

### 4.5.2 DIFFSPLAT 3D GENERATION

**Multi-view Manners** Two popular multi-view manners are explored in this work, yielding similar results in text-conditioned 3D generation, as shown in Table 5. View-concat manner performs better in single image-conditioned generation due to the dense attention among conditional image latents and splat latents. Although the spatial-concat method can also achieve dense attention by concatenating image latents along the spatial dimension, it requires additional padding, leading to increased computational costs. Thus, we prefer the view-concat manner in this work for its flexibility in accommodating varying numbers of viewpoints and conditioning.

**Training Objectives** DIFFSPLAT can perform well with merely the regular diffusion loss by setting $\lambda_{\text{render}} = 0$ given high-quality splat latents. However, as shown in Table 5 and Figure 6, the proposed 3D rendering loss $\mathcal{L}_{\text{render}}$ can further boost both the aesthetic quality and geometric structure of generated content. Aesthetic appeal and textured details may contributed by the perceptual loss, and fewer translucent artifacts are achieved through the mask loss described in Equation 1.

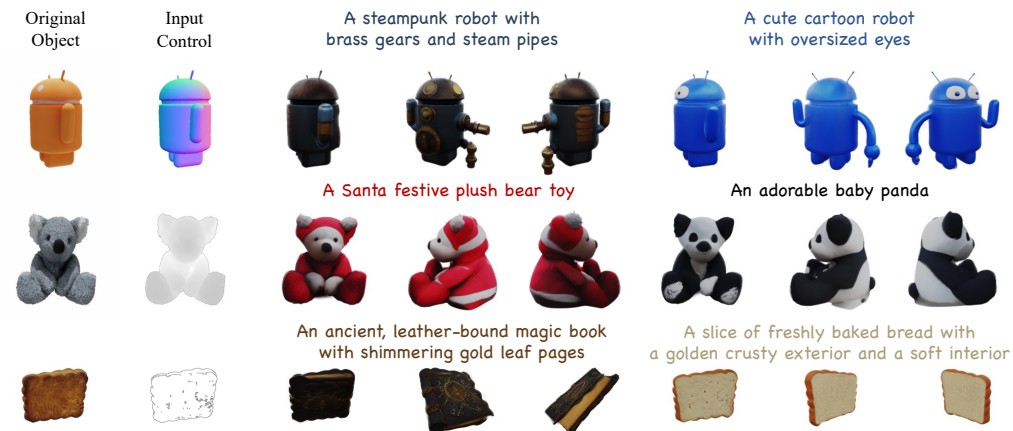

Figure 5: **Controllable Generation**. ControlNet can seamlessly adapt to DIFFSPLAT for controllable text-to-3D generation in various formats, such as normal and depth maps, and Canny edges.

Table 3: Ablation study of inputs for structured splat reconstruction.

|  | ↑ PSNR | ↑ SSIM | ↓ LPIPS | #Param. |
|---|---|---|---|---|
| LGM | 26.48 | 0.892 | 0.077 | 415M |
| GRM | 28.04 | 0.959 | 0.031 | 179M |
| RGB | 27.43 | 0.956 | 0.041 | 42M |
| +Normal | 28.89 | 0.957 | 0.033 | 42M |
| +Coord. | 29.87 | 0.961 | 0.028 | 42M |
| Ours | **30.09** | **0.963** | **0.027** | 42M |

Table 4: Ablation study for Gaussian splat property auto-encoding strategies.

|  | ↑ PSNR | ↑ SSIM | ↓ LPIPS |
|---|---|---|---|
| Frozen VAE | 8.64 | 0.833 | 0.566 |
| Frozen $E_{\phi_e}$ | 24.73 | 0.927 | 0.065 |
| w/o $\mathcal{L}_{\text{render}}$ | 26.07 | 0.937 | 0.080 |
| Ours (SD1.5) | 30.33 | 0.966 | 0.033 |
| Ours (SDXL) | 30.30 | 0.964 | 0.031 |
| Ours (SD3) | **31.08** | **0.978** | **0.025** |

DIFFSPLAT also produces reasonable results only with the rendering loss by setting $\lambda_{\text{diff}} = 0$. However, the training process becomes unstable and slow to converge, and gets over-saturated results.

**Base Text-to-image Diffusion Models**    Various popular open-source large text-to-image diffusion models are investigated in this work, including SD1.5 (Rombach et al., 2022), SDXL (Podell et al., 2024), PixArt-$\alpha$ (Chen et al., 2024c), PixArt-$\Sigma$ (Chen et al., 2024b) and SD3 (Esser et al., 2024), featuring different model sizes, backbone networks, noise scheduling, and sampling strategies. With the advancements in base models, DIFFSPLAT consistently benefits in both text- and image-conditioned tasks, indicating that the proposed method effectively leverages priors from pretrained models and stands as a promising approach for scaling 3D generation within the thriving image community.

### 4.5.3   INTERPRETING SPLAT LATENTS

To understand the feasibility of generating Gaussian splat properties through fine-tuning image diffusion models, we visualize splat latents and their corresponding properties to interpret the mechanisms behind DIFFSPLAT. As shown in Figure 7, auto-encoded Gaussian splat properties are presented as RGB or grayscale images. For rotations $\mathbf{r} \in \mathbb{R}^4$, the first three channels and the last one are visualized individually. Splat latents encoded by a fine-tuned VAE are decoded by the original **image** VAE. Gaussian splat property grids are analogous to natural images, reflecting the hue and edge properties of 3D objects. Decoded splat latents from the image VAE can be interpreted as the original objects in a "special style" or illuminated in a "special environment light", featuring a cyan light at a distance and a rufous light nearby. Image diffu-



Figure 7: **Splat Latents Visualization**. 3DGS properties are structured in grids. "Decoded GS" shows the splat latents decoded by an **image** diffusion VAE.

sion models are essentially fine-tuned to learn this special style, indicating that the input latents are splat latents, which enables the repurposing of image diffusion models to generate 3DGS.

Table 5: Ablation study of DIFFSPLAT design choices.

| | T3Bench-300 | | | GSO-300 | | |
|---|---|---|---|---|---|---|
| | ↑ CLIP Sim.$_\%$ | ↑ CLIP R-Prec.$_\%$ | ↑ ImageReward | ↑ PSNR | ↑ SSIM | ↓ LPIPS |
| *Multi-view Manner* | | | | | | |
| Spatial-concat | **28.74**$_{\pm0.08}$ | 55.92$_{\pm1.19}$ | -1.224$_{\pm0.023}$ | 21.61$_{\pm6.36}$ | 0.875$_{\pm0.086}$ | 0.121$_{\pm0.092}$ |
| View-concat | 28.66$_{\pm0.14}$ | **58.92**$_{\pm2.30}$ | **-1.201**$_{\pm0.031}$ | **22.58**$_{\pm6.05}$ | **0.885**$_{\pm0.082}$ | **0.117**$_{\pm0.085}$ |
| *Training Objective* | | | | | | |
| w/o $\mathcal{L}_{\text{diff}}$ | 27.72$_{\pm0.09}$ | 50.83$_{\pm1.91}$ | -1.436$_{\pm0.033}$ | 17.16$_{\pm3.20}$ | 0.794$_{\pm0.073}$ | 0.267$_{\pm0.132}$ |
| w/o $\mathcal{L}_{\text{render}}$ | 28.54$_{\pm0.22}$ | 56.42$_{\pm2.27}$ | -1.219$_{\pm0.049}$ | 22.30$_{\pm6.33}$ | 0.883$_{\pm0.085}$ | 0.125$_{\pm0.086}$ |
| $\mathcal{L}_{\text{render}} + \mathcal{L}_{\text{diff}}$ | **28.66**$_{\pm0.14}$ | **58.92**$_{\pm2.30}$ | **-1.201**$_{\pm0.031}$ | **22.58**$_{\pm6.05}$ | **0.885**$_{\pm0.082}$ | **0.117**$_{\pm0.085}$ |
| *Base Model (#Param.)* | | | | | | |
| SD1.5 (0.86B) | 28.66$_{\pm0.14}$ | 58.92$_{\pm2.30}$ | -1.201$_{\pm0.031}$ | 22.58$_{\pm6.05}$ | 0.885$_{\pm0.082}$ | 0.117$_{\pm0.085}$ |
| SDXL (2.6B) | 29.70$_{\pm0.22}$ | 63.00$_{\pm1.93}$ | -0.804$_{\pm0.038}$ | 22.65$_{\pm5.84}$ | 0.887$_{\pm0.084}$ | 0.115$_{\pm0.089}$ |
| PixArt-$\alpha$ (0.61B) | 29.10$_{\pm0.07}$ | 59.01$_{\pm1.26}$ | -1.052$_{\pm0.028}$ | 22.81$_{\pm5.90}$ | 0.884$_{\pm0.078}$ | 0.108$_{\pm0.080}$ |
| PixArt-$\Sigma$ (0.61B) | 29.75$_{\pm0.06}$ | 64.83$_{\pm0.50}$ | -0.834$_{\pm0.017}$ | 22.85$_{\pm5.75}$ | 0.890$_{\pm0.077}$ | **0.105**$_{\pm0.082}$ |
| SD3-medium (2.0B) | **30.15**$_{\pm0.05}$ | **68.08**$_{\pm0.72}$ | **-0.685**$_{\pm0.043}$ | **22.91**$_{\pm5.54}$ | **0.892**$_{\pm0.094}$ | 0.107$_{\pm0.093}$ |

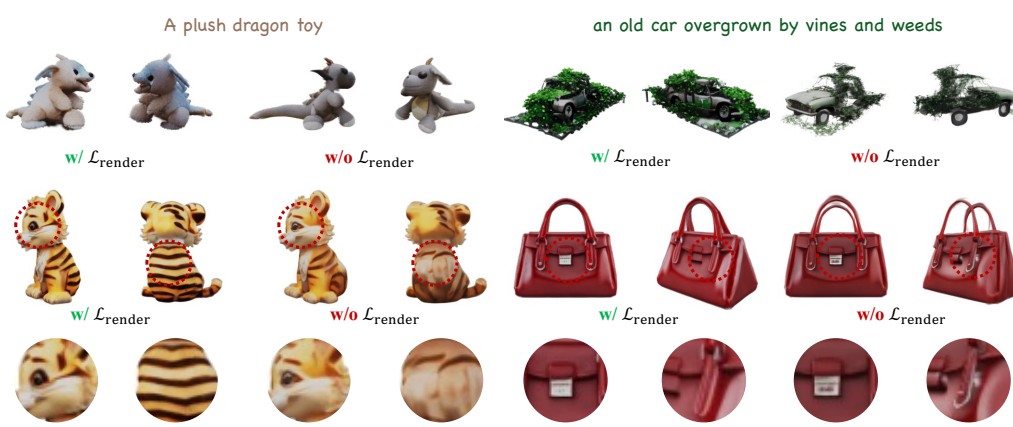

Figure 6: **Ablation of $\mathcal{L}_{\text{render}}$.** Both text- (1st row) and image-conditioned (2nd row) DIFFSPLAT with $\mathcal{L}_{\text{render}}$ produces more aesthetic and textured 3D content with fewer translucent floaters.

## 5 CONCLUSION

In this work, we present a novel diffusion-based 3D generation framework, DIFFSPLAT. It distinguishes from previous 3D generative methods by effectively leveraging web-scale 2D priors while maintaining 3D coherence in a unified model. Various large text-to-image diffusion models are fine-tuned to directly generate 3D Gaussian splat properties with both diffusion loss and 3D rendering loss. Thus, DIFFSPLAT benefits from the rapid developments in the image community, facilitating the integration of advanced techniques for image generation into the 3D realm. It positions DIFFSPLAT a promising 3D generative method utilizing abundant 2D priors and merely multi-view supervision. We hope this work can provide a new solution for 3D content creation.

**Limitations and Future Work** Although DIFFSPLAT delivers decent results, the conversion of its 3DGS representation to high-quality mesh remains an unsolved problem. Simultaneously generating splat latents from more viewpoints, increasing the supervision resolution, and integrating physical-based material properties could further enhance the quality of generated 3D Gaussian primitives. Numerous advanced techniques developed for image diffusion models can be adopted for 3D generation within the proposed framework, which remains underexplored, such as personalization, few-step distillation, and feedback learning to align with human preferences. Moreover, we only utilize rendered multi-view datasets in this work, which does not fully exploit the scalability potential of the proposed method. By leveraging its image compatibility and 2D supervision-only characteristic, massive real-world video datasets could further unlock the capability of DIFFSPLAT.

ACKNOWLEDGMENT

This work is supported by a grant from ByteDance (No. CT20240607105793) and an internal grant of Peking University (2024JK28).

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

## A IMPLEMENTATION DETAILS

**Reproducibility** We provide comprehensive implementation details in this section to facilitate the reproducibility of our work. Our code and models are publicly available at `https://chenguolin.github.io/projects/DiffSplat`.

**Training** For Gaussian splat grid reconstruction, we train a lightweight `12`-layer and `8`-head Transformer encoder (Vaswani et al., 2017) with `512` attention dimensions and a patch size of `8`, whose parameter size is only `42M` and `9.9%`∼`23%` of previous methods (Tang et al., 2024; Xu et al., 2024c; Zhang et al., 2024c). $s_{min}$ and $s_{max}$ are set to `5e-4` and `2e-2` respectively to represent fine-grain details. The input views $V_{in} = 4$ are evenly distributed and rendering views $V = 8$ include 4 other random viewpoints. All weighting terms are set to `1`. The probability of applying the rendering loss starts at `0` and is set to `1` later for training efficiency. All experiments are conducted at the $256 \times 256$ resolution in this work. Training batch size for reconstruction and auto-encoding is `64` in total across up to `16` A100 GPUs with gradient accumulation and the peak learning rate of `4e-4`. For diffusion models, the batch size and peak learning rate are `128` and `1e-4` respectively. AdamW optimizer (Loshchilov & Hutter, 2018) with weight decay and cosine learning rate scheduler (Loshchilov & Hutter, 2016) with linear warm-up are adopted for parameter optimization.

**Inference** For diffusion-based models (SD1.5 (Rombach et al., 2022), SDXL (Podell et al., 2024), PixArt-$\alpha$ (Chen et al., 2024c) and PixArt-$\Sigma$ (Chen et al., 2024b)), the DPM-Solver++ (Lu et al., 2022a;b) ODE solver with 20 inference steps is adopted following Chen et al. (2024c;b). The flow-based model, i.e., SD3 (Esser et al., 2024) uses the original flow matching Euler ODE solver (Lipman et al., 2023) with 28 steps, consistent with its original configuration. In the text-conditioned generation, classifier-free guidance (Ho & Salimans, 2021) scales for each model are the same with their default values: `7.5` for SD1.5, `5` for SDXL, `4.5` for PixArt-$\alpha$ and PixArt-$\Sigma$, and `7` for SD3. In the image-conditioned generation, all models are fine-tuned to predict velocity (Salimans & Ho, 2022; Shi et al., 2023), and their guidance scales are all set to `2`. Runtime for DIFFSPLAT to generate a single 3D object on an A100 GPU is only about **1∼2 seconds** with half precision.

**Cost** Notably, with 2D generative priors, DIFFSPLAT only takes about **3 days on 8 A100 GPUs** to generate decent results with `fp16` mixed precision, which is much more training-efficient than previous 3D generative models, such as DMV3D (Xu et al., 2024d) (128 A100 × 7 days), CLAY (Zhang et al., 2024d) (256 A800 × 15 days) and 3DTopia-XL (Hong et al., 2024a) (128 A100 × 14 days).

## B MORE VISUALIZATION RESULTS

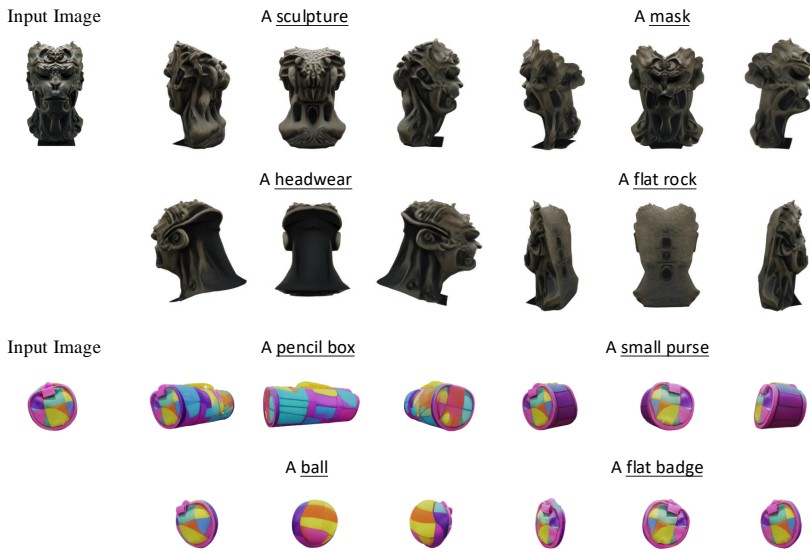

Figure 8: **Controllable Generation with Multi-modal Conditions**. DIFFSPLAT can effectively utilize both text and image conditions for single-view reconstruction with text understanding.

Text Prompt | Generated 3DGS

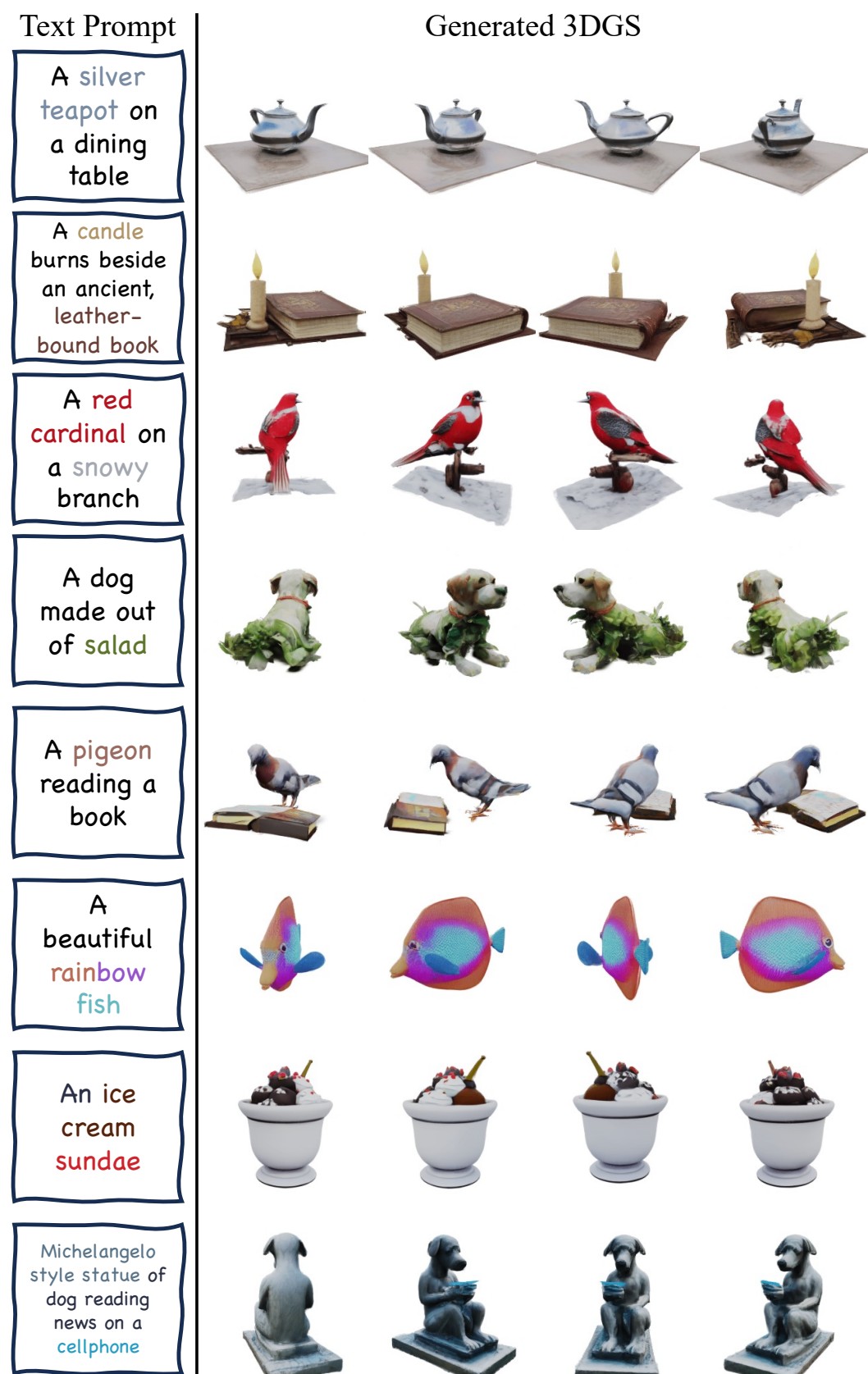

Figure 9: More results of text-conditioned DIFFSPLAT.

| Text Prompt | Generated 3DGS |
|---|---|

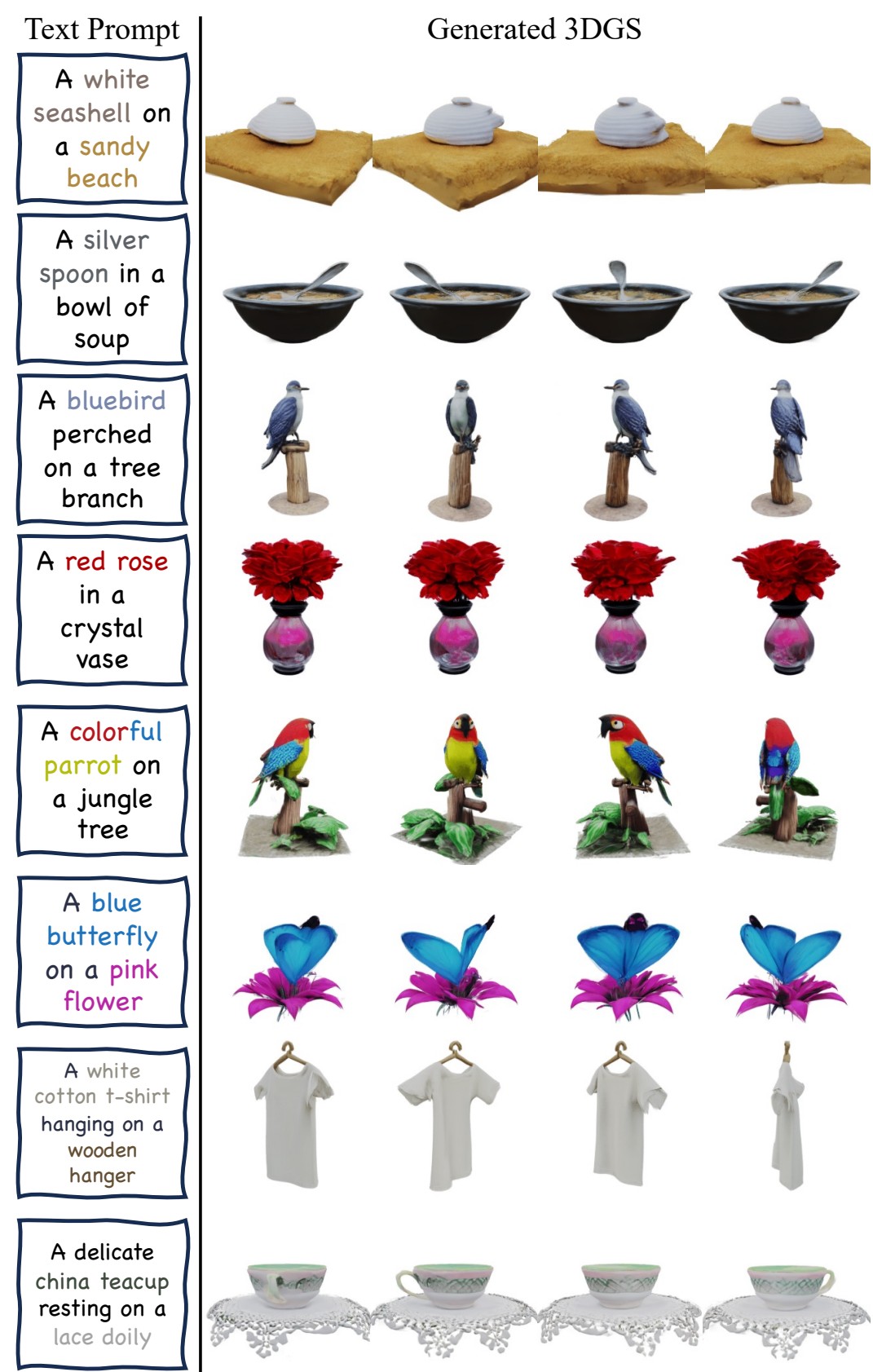

A white seashell on a sandy beach

A silver spoon in a bowl of soup

A bluebird perched on a tree branch

A red rose in a crystal vase

A colorful parrot on a jungle tree

A blue butterfly on a pink flower

A white cotton t-shirt hanging on a wooden hanger

A delicate china teacup resting on a lace doily

Figure 10: More results of text-conditioned DIFFSPLAT.

Text Prompt | Generated 3DGS

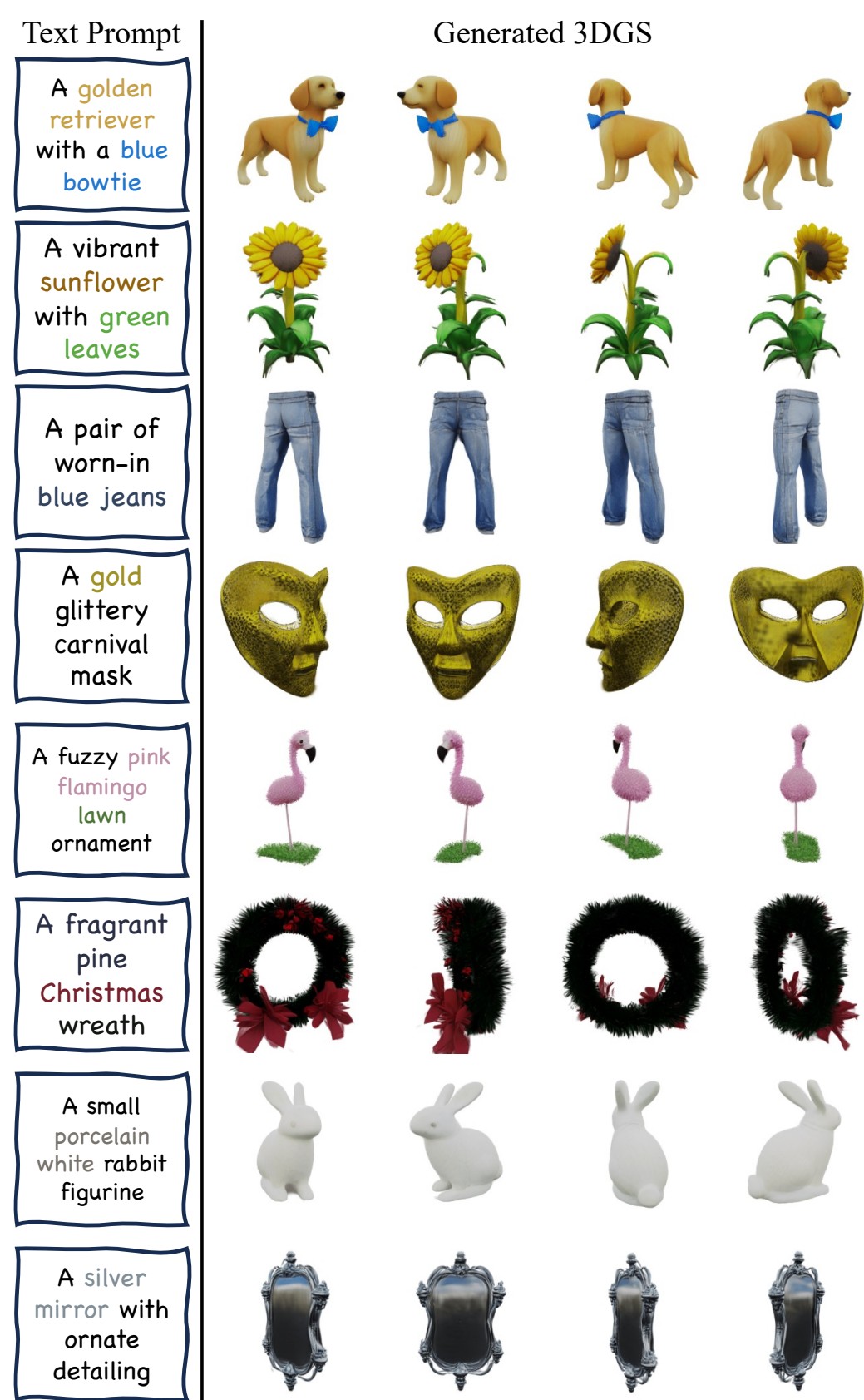

Figure 11: More results of text-conditioned DIFFSPLAT.

Input Image    Generated 3DGS

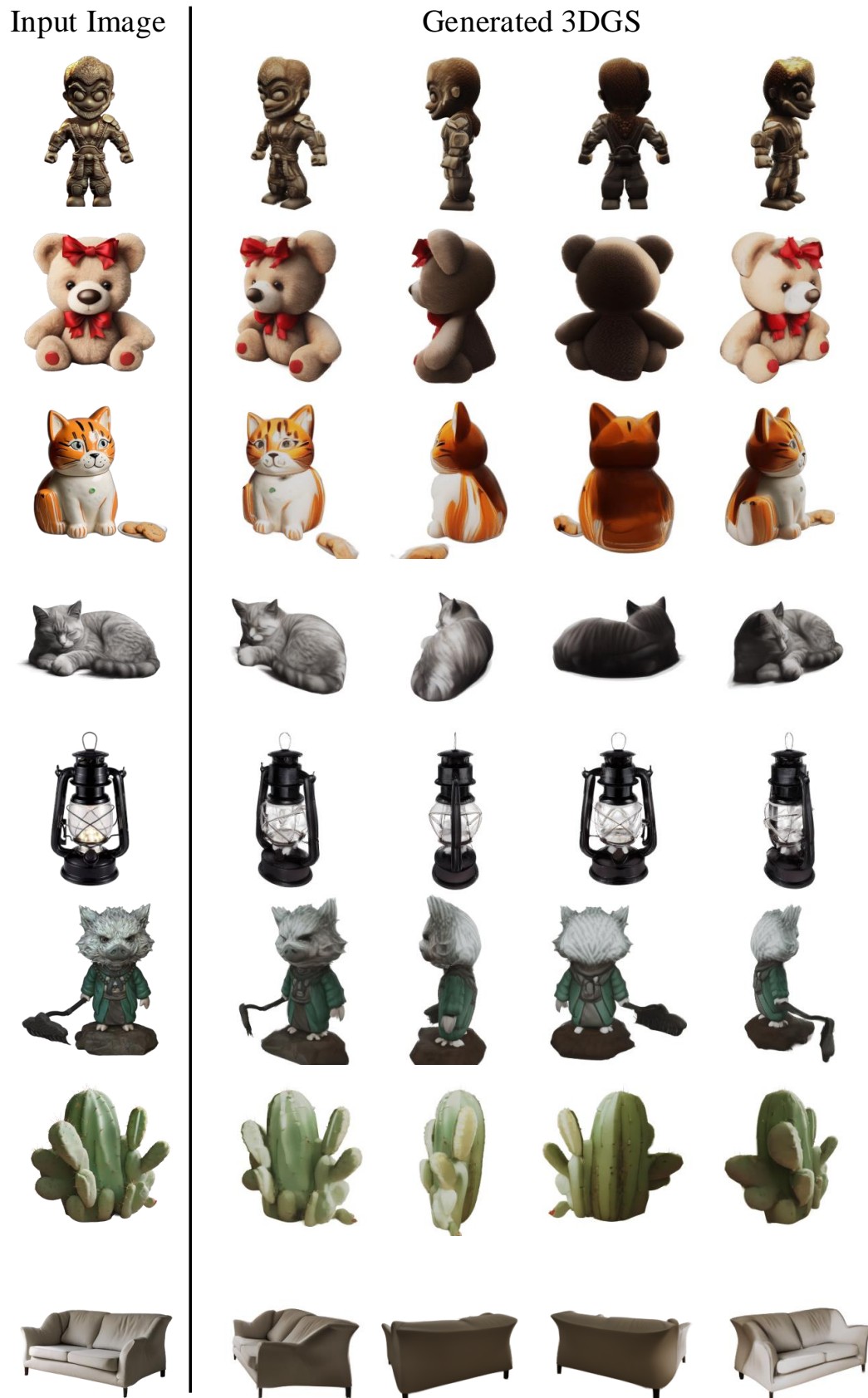

Figure 12: More results of image-conditioned DIFFSPLAT.

Input Image  Generated 3DGS

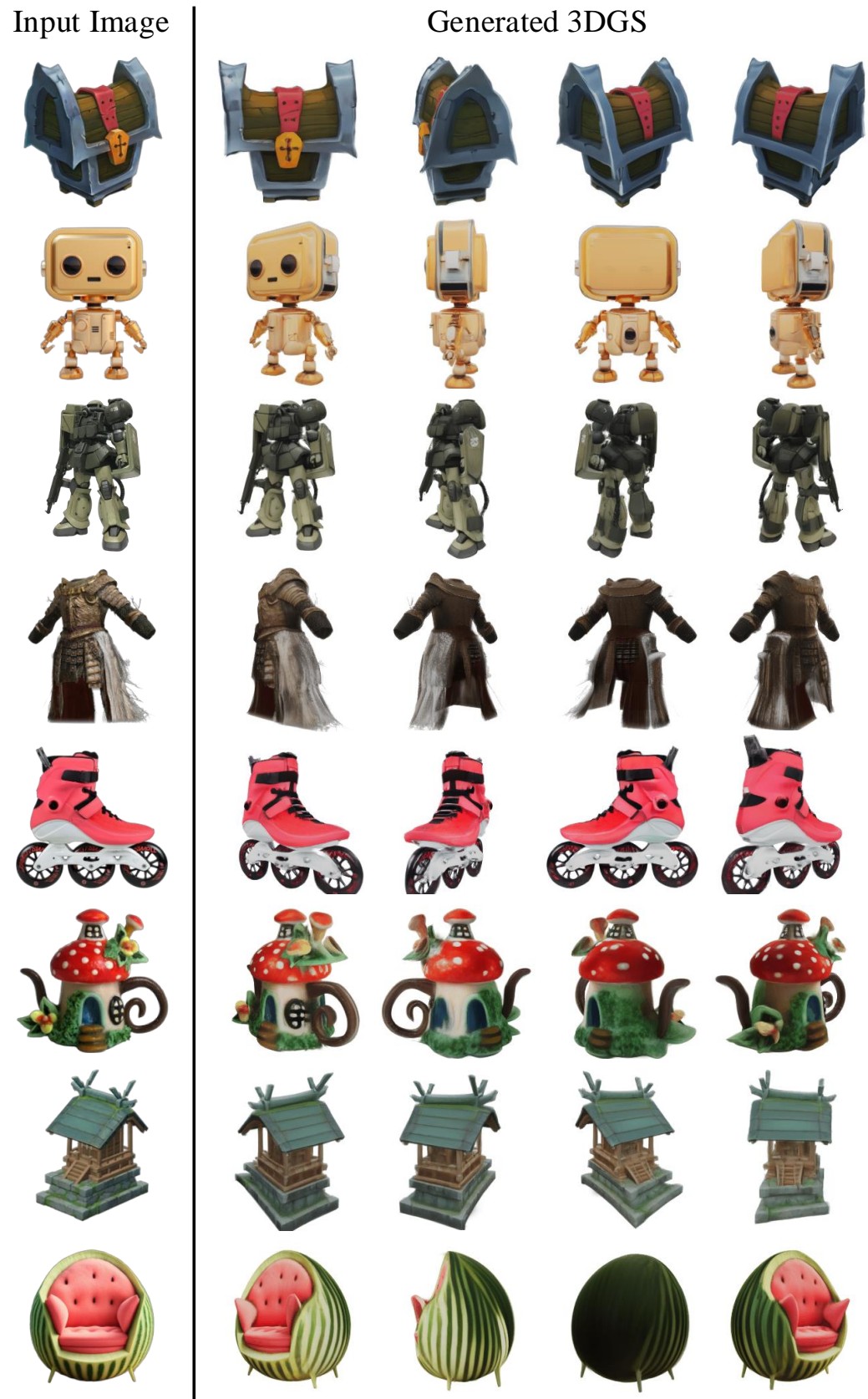

Figure 13: More results of image-conditioned DIFFSPLAT.

Input Image                              Generated 3DGS

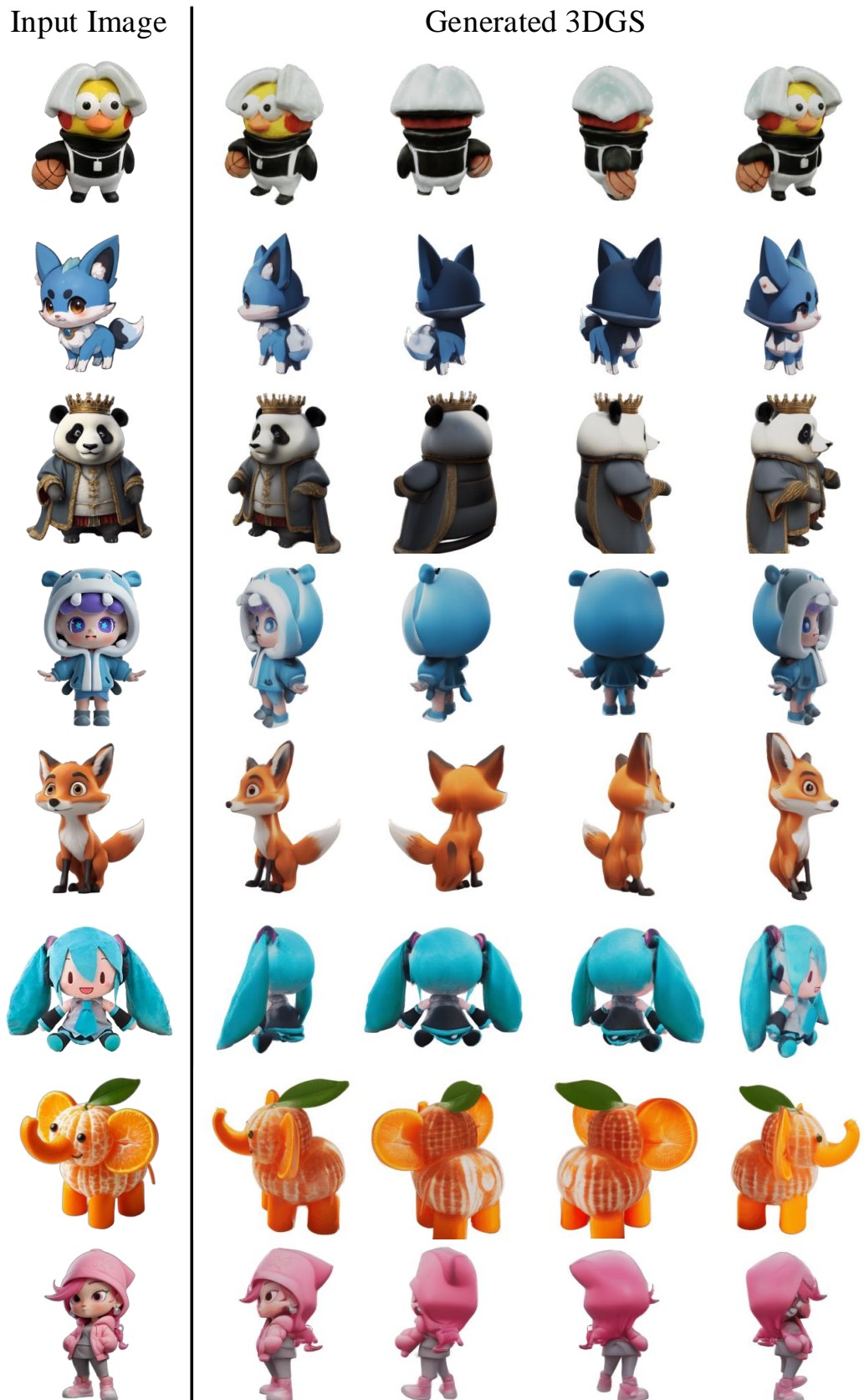

Figure 14: More results of image-conditioned DIFFSPLAT.

