# OpenReview forum: "DiffSplat: Repurposing Image Diffusion Models for Scalable Gaussian Splat Generation"
_ICLR.cc/2025/Conference — ICLR 2025 Poster_

### Official Review · Reviewer_L5NX · 2024-10-20

**Soundness:** 3
**Presentation:** 3
**Contribution:** 3
**Rating:** 8
**Confidence:** 5

**Summary:**

This paper proposes DiffGS, a novel 3D generative framework that leverages large-scale text-to-image diffusion models to directly generate 3D Gaussian splats.  It first introduces a lightweight, generalizable 3DGS network that regresses per-pixel Gaussians. Then, it fine-tunes existing VAEs to encode these per-pixel Gaussians into Gaussian latents. Finally, the large-scale text-to-image diffusion models are fine-tuned to predict the noise injected in Gaussian latents. Extensive experiments show that the proposed method achieves high-quality novel view synthesis results compared to baselines.

**Strengths:**

* The proposed method uses fewer parameters and achieves better novel view quality compared to baselines.

* A framework that generates 3DGS from noise is interesting.

* The paper is structured well, making it easy to understand and follow.

**Weaknesses:**

I think there is no critical weakness but there are some unclear components in this study:

1. The authors claim that DiffGS leverages large 2D diffusion priors and maintains 3D consistency through the 3D rendering loss. However, the reported results focus only on novel view synthesis and text similarity scores. It would be better to include evaluations of 3D consistency, similar to the COLMAP-based evaluation used in SyncDreamer [1] or the optical flow-based evaluation used in HarmonyView [2].

2. At the inference stage, how is the input camera pose obtained? Is it derived from the training dataset?

3. Although the Gaussian latents are already aligned in the 3D representation, is there still a need for Plücker ray embeddings? What would happen if ray embeddings were not used?

[1] Liu et al., Syncdreamer: Generating multiview-consistent images from a single-view image, ICLR 2024.

[2] Woo et al., Harmonyview: Harmonizing consistency and diversity in one-image-to-3d, CVPR 2024.

**Questions:**

See weakness

---

> ### Author Response · Authors · 2024-11-21
> **Response to Reviewer L5NX - Part 1**
>
> We sincerely appreciate the reviewer's insightful and valuable feedback. It is really inspiring to know that you consider our work to have no critical weaknesses and find the framework interesting, more parameter-efficient than the baselines, and well-structured. Below are our clarifications regarding your concerns. We value your input and would welcome any follow-up questions or suggestions you may have. If you find our responses helpful, we sincerely hope the reviewer could kindly consider raising the score accordingly.
>
> **1. It would be better to include evaluations of 3D consistency, such as COLMAP-based and optical flow-based evaluations.**
>
> Thank you for your suggestion regarding the evaluation of 3D consistency. We would like to note that the generated content in our work **is already represented in 3D (i.e., 3DGS). This distinguishes it from multi-view image generation studies**, such as the impressive works SyncDreamer [1] and HarmonyView [2]. That is why RealFusion [3], which directly optimizes a 3D NeRF, achieves a significantly higher COLMAP score (i.e., number of reconstructed points) though it has lower NVS metrics, as shown in Table 1 of SyncDreamer [1], and is not evaluated for the optical flow-based score in Table 2 of HarmonyView [2]. However, inconsistencies arising from multi-view input images can lead to poor reconstruction results with floating artifacts. Therefore, it is feasible to evaluate 3D consistency using metrics such as CLIP score, ImageReward, PSNR, and LPIPS.
>
> COLMAP-based evaluations as SyncDreamer [1] for the rendering results from our method are conducted as follows.
>
> |                | Ours (text) | Ours (image) | Realfusion | SyncDreamer | Zero123 |
> |----------------|-------------|--------------|------------|-------------|---------|
> | $\uparrow$ COLMAP #Points | 6296        | 6637         | 4010       | 1123        | 95      |
>
> Thank you again for the great suggestion of using a COLMAP-based method to evaluate 3D consistency. We will add these quantitative results in the revision of the manuscript.
>
> **2. How is the input camera pose obtained at the inference stage?**
>
> We apologize for the lack of clarity in the manuscript. The input camera pose during inference **is not derived from the training dataset**.
>
> During training, camera poses in the used training dataset (i.e., G-Objaverse) are randomly sampled for each object. While during inference:
>
> - For the text-to-3D model, the camera elevation can be selected randomly, and we use a default elevation of `10°` for evaluation. The azimuth angles are evenly distributed at `0°`, `90°`, `180°`, and `270°`.
>
> - For the image-to-3D model, the azimuth angles are also evenly distributed, with `0°` corresponding to the viewpoint of the input image. The elevation is determined by the input image and is set as an input argument. We use `10°` as the default elevation for evaluation, as this setting performs well in most cases. We found that the **rendering loss helps the models be less sensitive to input elevations**. Additionally, we can either use an **off-the-shelf elevation estimation module** or design a more sophisticated one to better automate the image-conditioned generative pipeline, similar to One-2-3-45 [7] and Era3D [8].
>
> Thank you for mentioning that and we will include details about camera pose in the revision of the manuscript.

---

> ### Author Response · Authors · 2024-11-21
> **Response to Reviewer L5NX - Part 2**
>
> **3. Is there still a need for Plücker ray embeddings for Gaussian latents?**
>
> That's a very good question! Thank you for bringing it up to the discussion. As Gaussian latents are compressed from Gaussian attributes that indicate 3D awareness, it appears that Plücker ray embeddings are redundant for Gaussian latents. However, in our work, the encoded Gaussian attribute related to the position is **depth**, rather than **XYZ location** as in LGM. Depth values from different views can't be distinguished and fail to offer view awareness. Essentially, using `depth + Plücker` implies the actual XYZ location, and Plücker ray embeddings are indeed redundant in our reconstruction model, which utilizes additional coordinate maps are inputs. But Plücker ray embeddings are still necessary for Gaussian latents.
>
> We also conducted an ablation study on Plücker ray embeddings with the SD1.5-based text-to-3D model as follows. An SD1.5-based text-to-3D model without Plücker concatenation is trained from scratch for comparison. The experimental results demonstrate that the 3D consistency between different views was compromised without the use of Plücker ray embeddings.
>
> |                   |CLIP Sim. (\%) $\uparrow$  |CLIP R-Prec. (\%) $\uparrow$  | ImageReward $\uparrow$  |
> |-------------------|----------------|-------------------|-------------|
> | w/o Plücker       | 29.40          | 72.00             | -1.169      |
> | w/ Plücker (Ours) | 30.51          | 79.00             | -0.553      |
>
>
> Thank you again for mentioning that and we will add this discussion about Plücker ray embeddings in the revision of the manuscript.
>
> ---
>
> [1] Liu et al., Syncdreamer: Generating multiview-consistent images from a single-view image, ICLR 2024.
>
> [2] Woo et al., Harmonyview: Harmonizing consistency and diversity in one-image-to-3d, CVPR 2024.
>
> [3] Melas-Kyriazi et al., RealFusion: 360° Reconstruction of Any Object from a Single Image, CVPR 2023.
>
> [4] Shi et al., MVDream: Multi-view Diffusion for 3D Generation, ICLR 2024.
>
> [5] Shi et al., Zero123++: a Single Image to Consistent Multi-view Diffusion Base Model, arXiv 2024.
>
> [6] Leroy et al., Grounding Image Matching in 3D with MASt3R, ECCV 2024.
>
> [7] Liu et al., One-2-3-45: Any Single Image to 3D Mesh in 45
> Seconds without Per-Shape Optimization, NeurIPS 2023.
>
> [8] Li et al., Era3D: High-Resolution Multiview Diffusion
> using Efficient Row-wise Attention, NeurIPS 2024.

---

> > ### Comment · Reviewer_L5NX · 2024-11-25
> > **Response to the author comment**
> >
> > Thank you for the response. I believe this study introduces a novel approach to generating 3D Gaussian splats, and the author has addressed all concerns. Therefore, I will raise my rating to 8.

---

### Official Review · Reviewer_2Rvk · 2024-10-29

**Soundness:** 4
**Presentation:** 4
**Contribution:** 4
**Rating:** 8
**Confidence:** 5

**Summary:**

This paper, DIFFGS, proposes a new 3D generative framework designed to improve 3D content generation, addressing challenges like limited high-quality datasets and inconsistencies from 2D multi-view generation. Unlike other models, DIFFGS harnesses large-scale text-to-image diffusion models, leveraging 2D priors while ensuring 3D consistency. It introduces a lightweight reconstruction model to create multi-view Gaussian grids, enabling scalable data curation. Alongside traditional diffusion loss, DIFFGS uses a 3D rendering loss to enhance 3D coherence across views. This framework is compatible with image diffusion models, allowing seamless integration of image-generation techniques into 3D applications. Experiments show DIFFGS excels in text- and image-conditioned tasks, with ablations confirming the importance of each component. Code and models will be publicly available.

**Strengths:**

1. This paper presents a novel paradigm of 3D generation, leveraging strong 2D pretrained T2I models. The proposed method first encodes the given object into multi-view splatter images, fine-tunes the 2D VAEs to the 12xHxW domain, then train the multi-view splatter images diffusion models with both diffusion loss and rendering loss. Both view-concat  and spatial-concat versions perform reasonably well.
2. The comparison experiment is sound, which included the most recent methods (LN3Diff, 3DTopia-XL) as the strong baselines, which greatly demonstrates the effectiveness of the proposed method.
3. The proposed method can be well integrated into existing 2D diffusion frameworks, e.g., introducing ControlNet into the 3D generation domain.
4. Overall, I like this paper very much and strongly lean to accept it as the ICLR main conference paper.

**Weaknesses:**

Whether representing the given 3D object with multi-view splatter images leads to inconsistency, like in LGM? Since the Gaussians have overlap between different regions, and some “ghosting artifacts” are very common especially in very thin objects (such as axes).

**Questions:**

1. When fine-tuning the 2D VAEs, have you fixed some parameters or you fine-tune the whole VAE model? Since if fine-tuning the whole 2D VAE, how to stay connected with the pre-trained diffusion latent space?
2. Does your stage-1 reconstruction model for data curation itself serve as a LGM/GRM-like model, since it consumes multi-view RGB-N-XYZ renderings into the pixel-aligned Gaussians.
3. Comparison with amortised SDS-based model (e.g., LATTE3D: Large-scale Amortized Text-To-Enhanced3D Synthesis). They also provide a way to leverage pre-trained 2D diffusion models for 3D generation.
4. One disadvantage of the proposed solution is a lack of 3D latent space, and the generation of interior regions of the 3D object. Is these issues solvable by augmenting the proposed framework?
5. Compared to training diffusion on the 3D latent space direction (e.g., 3DTopia and LN3Diff), what are the limitations of the proposed method?

---

> ### Author Response · Authors · 2024-11-21
> **Response to Reviewer 2Rvk - Part 1**
>
> We sincerely appreciate the reviewer's insightful and valuable feedback. It is really inspiring to know that you like our work, lean strongly toward acceptance, and view it as a novel paradigm in 3D generation. We are also delighted to hear that you found our experiments sound and recognized the proposed method's compatibility with 2D diffusion models. Below are our clarifications regarding your concerns. We value your input and would welcome any follow-up questions or suggestions you may have.
>
> **1. Does representing a 3D object with multi-view splatter images lead to inconsistencies, especially for very thin objects like axes?**
>
> That's a very good question! Thank you for bringing it up to the discussion. Note that the generated content in our work is already represented in 3D (i.e., 3DGS). This distinguishes it from multi-view image generation studies, so our generated results are already fully 3D consistent. However, it could cause floating artifacts if there are ``inconsistencies'' among multi-view splatter images. Fortunately, two kinds of designs help our methods improve coherence among generated latents:
>
> - **Gaussian latents (implicit 3D awareness)**: Besides RGB colors that resemble image pixels, Gaussian latents are also encoded with attributes such as depth, rotation and scale, which indicate 3D awareness. By generating these 3D attributes as a whole, our generative model implicitly gains a better understanding of 3D space and enhances the consistency across splatter images.
>
> - **Rendering loss (explicit 3D awareness)**: In addition to the standard diffusion loss, thanks to the design that a renderable 3D representation (i.e., splatter image) is processed during the diffusion process, we introduce rendering loss to explicitly promote model output appears plausible from multiple viewpoints in 3D space.
>
> Visualizations of very thin objects (such as axes) generated by our methods are provided in [https://iclr2025-paper1469.github.io/index.html#thin](https://iclr2025-paper1469.github.io/index.html#thin).
>
> Nevertheless, we acknowledge that there are limitations to the multi-view splatter images we used. More discussion and potential solutions are provided in the 6th (last) question. Thank you for highlighting this issue, and we will include this discussion in the revised manuscript.
>
> **2. Have you fixed some parameters when fine-tuning the 2D VAEs? How to stay connected with the pretrained diffusion latent space if tuning the whole 2D VAE?**
>
> No, we do **not fix any parameters when fine-tuning the 2D VAEs**. It's also surprising to us that, even when starting with the simplest experimental settings (i.e., no KL regularization, no GAN loss, and no sophisticated parameter selection), it still works well and stays connected with the pretrained diffusion latent space to a certain extent, as shown in **Figure 7** of the manuscript.
>
> Thank you for bringing it up to the discussion. The connection between Gaussian and pixel latent spaces is indeed an intriguing topic for further exploration. We will include this discussion in the ``Limitations and Future Work'' section in the revised manuscript.
>
> **3. Does the stage-1 reconstruction model for data curation itself serve as an LGM/GRM-like model?**
>
> Yes, the stage-1 reconstruction model is essentially an LGM/GRM-like model. However, there are **three main differences**:
>
> - **Normalized attributes**: In our work, all Gaussian attributes are designed to be well-normalized within the range of [0, 1], which provides a suitable numerical scale that resembles pixel values for subsequent VAE and diffusion models. However, Gaussian attributes in LGM [1] and GRM [2] are not guaranteed to be normalized.
>
> - **Additional inputs (normal and coordinate maps)**: Besides multi-view RGB images, we also provide coordinate and normal maps as inputs to significantly enhance the quality of reconstructed Gaussians. This design stems from the fact that the stage-1 model is primarily served for data curation, where quality is paramount and inputs are not limited to RGB images, which differs from the requirements in LGM and GRM.
>
> - **Much fewer parameters**: Thanks to the aforementioned designs, we can achieve better reconstruction results with much fewer (9.9%~23%) parameters as shown in **Table 3** of the manuscript. The number of parameters in our stage-1 model is only **42M**, which is not considered as a ``large'' reconstruction model as LGM (415M), GRM (179M) and GS-LRM (300M) [3].

---

> ### Author Response · Authors · 2024-11-21
> **Response to Reviewer 2Rvk - Part 2**
>
> **4. Comparison with amortized SDS-based methods, such as LATTE3D.**
>
> We apologize for the lack of comparison with amortized SDS-based methods in the manuscript. Although we try our best to collect state-of-the-art text- and image-conditioned 3D generative models, some of them are not open-sourced for evaluation such as the mentioned amazing works ATT3D [4] and its follow-up LATTE3D [5]. It's interesting to note that the idea of amortized SDS may also be feasible for the proposed framework by replacing LPIPS in the rendering loss with an SDS-based loss. Moreover, the reconstruction pretraining and annealing trick proposed by LATTE3D may not be necessary in our framework, further simplifying amortized SDS-based methods.
>
> We try our best to provide a comparison between the proposed method and LATTE3D by running DreamFusion prompts provided on the project page of LATTE3D. Qualitative evaluations are provided in [https://iclr2025-paper1469.github.io/index.html#latte3d](https://iclr2025-paper1469.github.io/index.html#latte3d).
>
> We will include this discussion in the ``Limitations and Future Work'' section in the revised manuscript.
>
> **5. How to solve the lack of 3D latent space and the generation of interior regions of 3D objects?**
>
> We leave the discussion about solving the lack of 3D latent space to the 6th (last) question.
>
> Generating the interior regions of 3D objects is an intriguing yet often overlooked task in most 3D generative methods. To tackle this challenge, it is essential to prepare datasets that provide detailed interior regions for 3D objects and scenes, such as ShapeNet-Cars [6] for objects and 3D-FRONT [7] for scenes. Rendering-based 3D representations, such as NeRF and 3DGS, are not suitable for generating interior regions. Instead, numerous studies based on unsigned distance fields (UDF) can address this issue, such as CAP-UDF [8] and DUDF [9], which focus on reconstructing non-watertight surfaces from 3D point clouds, which is **beyond the scope of our work**. Another promising approach for generating interior regions involves assembling our generated objects from smaller components to form complex 3D content. We believe this direction holds significant potential for future research in 3D generation with detailed interior regions.
>
> **6. What are the limitations of the proposed method compared to 3D diffusion models with 3D latent space, such as 3DTopia and LN3Diff?**
>
> That is a great question! As the reviewer commented, the most significant limitation of multi-view splatter images is the lack of a 3D latent space. Thus, our work employs 3D-aware Gaussian latents and rendering loss to enhance the coherence of the generated content both implicitly and explicitly. However, as mentioned in the ``Limitation and Future Work'' section of the original manuscript, four evenly distributed views are insufficient to capture all the details of a 3D object. A true 3D latent space could significantly improve geometric precision and reduce floating artifacts.
>
> 3DTopia [10] and LN3Diff [11] utilize a triplane-based 3D latent space and train a diffusion model from scratch within this latent triplane. While the latent triplane offers a coherent 3D representation without overlap, which helps to eliminate floating artifacts, it is quite different from the latent spaces of pretrained image generative models that rely on views from above and below. This difference presents **challenges in utilizing 2D priors** and necessitates training from scratch. Qualitative and quantitative comparisons between these methods and ours can be found in Figure 3 and Table 1 of the original manuscript, demonstrating that our method achieves superior prompt alignment and visual quality, particularly for complex prompts.
>
> Fortunately, to address the lack of a 3D latent space, the proposed method can be **seamlessly augmented with a 3D VAE decoder**. Specifically, inspired by LRM [12] and CLAY [13], rather than using multi-view Gaussian latents as inputs for decoding Gaussian attributes, the generated latents can serve as keys and values to consistently influence queries. These queries could be implemented as learnable triplane tokens [12] or densely sampled positional embeddings within a 3D grid [13]. This approach retains all the advantages of the proposed method while incorporating a 3D latent space.  We believe it is a promising direction for future exploration.
>
> Thank you for bringing it up and we will add this discussion in the revision of the manuscript.

---

> ### Author Response · Authors · 2024-11-21
> **Response to Reviewer 2Rvk - Part 3**
>
> ---
>
> [1] Tang et al., LGM: Large Multi-View Gaussian Model for High-Resolution 3D Content Creation, ECCV 2024.
>
> [2] Xu et al., GRM: Large Gaussian Reconstruction Model for Efficient 3D Reconstruction and Generation, ECCV 2024.
>
> [3] Zhang et al., GS-LRM: Large Reconstruction Model for 3D Gaussian Splatting, ECCV 2024.
>
> [4] Lorraine et al., ATT3D: Amortized Text-to-3D Object Synthesis, ICCV 2023.
>
> [5] Xie et al., LATTE3D: Large-scale Amortized Text-To-Enhanced3D Synthesis, ECCV 2024.
>
> [6] Chang et al., ShapeNet: An Information-Rich 3D Model Repository, arXiv 2015.
>
> [7] Fu et al., 3D-FRONT: 3D Furnished Rooms with layOuts and semaNTics, ICCV 2021.
>
> [8] Zhou et al., CAP-UDF: Learning Unsigned Distance Functions Progressively from Raw Point Clouds with Consistency-Aware Field Optimization, NeurIPS 2022.
>
> [9] Fainstein et al., DUDF: Differentiable Unsigned Distance Fields with Hyperbolic Scaling, CVPR 2024.
>
> [10] Hong et al., 3DTopia: Large Text-to-3D Generation Model with Hybrid Diffusion Priors, arXiv 2024.
>
> [11] Lan et al., LN3Diff: Scalable Latent Neural Fields Diffusion for Speedy 3D Generation, ECCV 2024.
>
> [12] Hong et al., LRM: Large Reconstruction Model for Single Image to 3D, ICLR 2024.
>
> [13] Zhang et al., CLAY: A Controllable Large-scale Generative Model for Creating High-quality 3D Assets, TOG 2024.

---

### Official Review · Reviewer_i4mY · 2024-10-30

**Soundness:** 3
**Presentation:** 3
**Contribution:** 3
**Rating:** 6
**Confidence:** 3

**Summary:**

The authors propose a new diffusion-based multi-view generative models using 3D Gaussian Splatting rather than directly on image space. They consist of latent space which compress the gaussian representation to apply Latent Diffusion Model (LDM). Both diffusion loss and rendering loss are combined for better multi-view consistent plausible outputs.

**Strengths:**

1. The authors propose a way to combine 3DGS with latent diffusion model for generating better 3D models.
2. The proposed 3D generation pipeline seems to be both plausible and novel in that it parametrizes Gaussians and adopts specific VAE architectures to apply LDM on multiple Gaussian sets for 3DGS.
3. The authors well demonstrate both quantitative and qualitative results compared to competitive methods in image-to-3D and text-to-3D tasks.
4. Ablation study shows how each component contributes to the performance including omitting 3D rendering loss and 2D diffusion loss.

**Weaknesses:**

1. The contribution of the paper is on synthesizing conditional 3D models (text, single image) without per-object optimization using SDS such as DreamFusion, MVDream, and Wonder3D. I think the paper lacks some recent competitive methods including ATT3D [1] and Latte3D [2] which also aims to synthesize 3D models without per-object optimization.

2. There is no comparison for synthesizing time during inference. As the required time to get a single object is important, it can be also a weakness for the paper.

[1] Lorraine, Jonathan, et al. "Att3d: Amortized text-to-3d object synthesis." Proceedings of the IEEE/CVF International Conference on Computer Vision. 2023.

[2] Xie, Kevin, et al. "Latte3d: Large-scale amortized text-to-enhanced3d synthesis." European Conference on Computer Vision. 2024.

**Questions:**

1. Please compare the proposed methods with more 3D multi-view generative models such as Att3D [1] or Latte3D [2].
2. Please report rendering time of both single 3D model and multi-view 2D images with the rendered 3D model.
3. It would be good if the authors compare the 3D model quality of the proposed method with per-single 3D optimization models like GaussianDreamer [3].

[1] Lorraine, Jonathan, et al. "Att3d: Amortized text-to-3d object synthesis." Proceedings of the IEEE/CVF International Conference on Computer Vision. 2023.

[2] Xie, Kevin, et al. "Latte3d: Large-scale amortized text-to-enhanced3d synthesis." European Conference on Computer Vision. 2024.

[3] Yi, Taoran, et al. "Gaussiandreamer: Fast generation from text to 3d gaussians by bridging 2d and 3d diffusion models." Proceedings of the IEEE/CVF Conference on Computer Vision and Pattern Recognition. 2024.

---

> ### Author Response · Authors · 2024-11-21
> **Response to Reviewer i4mY**
>
> We sincerely appreciate the reviewer's insightful and valuable feedback. It is really inspiring to know that you regard our work as plausible and novel, and that our experiments and ablation studies sufficiently demonstrate the effectiveness of our methods. Below are our clarifications regarding your concerns. We value your input and would welcome any follow-up questions or suggestions you may have. If you find our responses helpful, we sincerely hope the reviewer could kindly consider raising the score accordingly.
>
> **1. Comparison with amortized SDS-based methods, such as LATTE3D and ATT3D.**
>
> We apologize for the lack of comparison with amortized SDS-based methods in the manuscript. Although we try our best to collect state-of-the-art text- and image-conditioned 3D generative models, some of them are not open-sourced for evaluation such as the mentioned amazing works ATT3D [1] and its follow-up LATTE3D [2]. It's interesting to note that the idea of amortized SDS may also be feasible for the proposed framework by replacing LPIPS in the rendering loss with an SDS-based loss. Moreover, the reconstruction pretraining and annealing trick proposed by LATTE3D may not be necessary in our framework, further simplifying amortized SDS-based methods.
>
> We try our best to provide a comparison between the proposed method and LATTE3D by running DreamFusion prompts provided on the project page of LATTE3D. Qualitative evaluations are provided in [https://iclr2025-paper1469.github.io/index.html#latte3d](https://iclr2025-paper1469.github.io/index.html#latte3d).
>
> We will include this discussion in the ``Limitations and Future Work'' section in the revised manuscript.
>
> **2. Comparison for synthesizing time during inference.**
>
> We have included the inference time of the proposed method in **Appendix A** of the submitted manuscript. It only takes about **1~2 seconds** for the proposed method to generate a single 3D content on an A100 GPU.
>
> We'd like to note that the model architectures of the proposed method are exactly the same as the corresponding image generative models (i.e., SD1.5, SDXL, PixArt-alpha, PixArt-Sigma and SD3-medium). So any acceleration techniques proposed to them, such as DeepCache [3], T-GATE [4] and Layer caching [5], can be directly applied to our models.
>
> We provide quantitative evaluations of synthesizing time during inference as follows. Our model architecture is similar to the multi-view image generation modules of LGM/GRM/LaRa/InstantMesh, but it operates at a smaller resolution.
>
> |     Text-to-3D     | Ours | GVGEN | LN3Diff | DIRECT-3D | 3DTopia | LGM | GRM |
> |--------------------|------|-------|---------|-----------|---------|-----|-----|
> | Inference Time (s) | 2    | 7     | 4       | 6         | 14      | 5   | 5   |
>
> |     Image-to-3D    | Ours | 3DTopia-XL | LN3Diff | LGM | GRM | LaRa | CRM | InstantMesh|
> |--------------------|------|------------|---------|-----|-----|------|-----|------------|
> | Inference Time (s) | 2    | 5          | 4       | 5   | 5   | 5    | 10  | 10         |
>
> We will include the inference speed evaluation in the revised manuscript.
>
> **3. Comparison with per-single 3D optimization models like GaussianDreamer.**
>
> We apologize for the lack of comparison with per-single 3D optimization (i.e., SDS-based) methods in the manuscript. We acknowledge that there are plenty of great pioneering works based on SDS for text- and image-conditioned 3D generation. However, due to space limitations, they are not discussed in the manuscript as they typically require several minutes or even hours to generate a single 3D object or scene, in contrast to the seconds required by the methods we included in the related work section. While SDS methods often yield more realistic and detailed results and are tailored to specific user cases through test-time optimization, they can serve as an upper bound or target goal for feed-forward 3D generative methods. Meanwhile, our generated results can be used to initialize SDS-based methods for further refinement, which is a promising direction for future exploration.
>
> We provide qualitative evaluations of the proposed method and SDS-based methods including GaussianDreamer [6] in [https://iclr2025-paper1469.github.io/index.html#sds](https://iclr2025-paper1469.github.io/index.html#sds).
>
> We will include this discussion in the ``Limitations and Future Work'' section in the revised manuscript.
>
> ---
>
> [1] Lorraine et al., ATT3D: Amortized Text-to-3D Object Synthesis, ICCV 2023.
>
> [2] Xie et al., LATTE3D: Large-scale Amortized Text-To-Enhanced3D Synthesis, ECCV 2024.
>
> [3] Ma et al., DeepCache: Accelerating Diffusion Models for Free, CVPR 2024.
>
> [4] Lin et al., Faster Diffusion via Temporal Attention Decomposition, arXiv 2024.
>
> [5] Ma et al., Learning-to-Cache: Accelerating Diffusion Transformer via Layer Caching, NeurIPS 2024.
>
> [6] Yi et al., GaussianDreamer: Fast Generation from Text to 3D Gaussian Splatting with Point Cloud Priors, CVPR 2024.

---

> > ### Comment · Reviewer_i4mY · 2024-11-22
> >
> > I really appreciate the authors' response.
> >
> > The results seem to be satisfactory and all the questions are resolved.
> >
> > Therefore, I'll keep my current score as 'marginally above the acceptance threshold' considering the novelty and the expected impact to the research field of this paper.

---

### Official Review · Reviewer_Xhe6 · 2024-11-04

**Soundness:** 2
**Presentation:** 3
**Contribution:** 3
**Rating:** 6
**Confidence:** 4

**Summary:**

This paper presents DiffGS, a new method to directly generate 3D Gaussians in a feed-forward manner. DiffGS is fine-tuned from image diffusion models with the 3D Gaussian latent. The experiments on both text and image condition 3D generation show that DiffGS can quickly generate reasonable 3D Gaussians with high-quality rendering quality.

**Strengths:**

1. DiffGS encodes the 3D Gaussian Splatting to latents and trains a diffusion model to generate the latents directly, which is efficient compared with the optimization-based methods. The 3D Gaussian Splatting also offers better rendering quality.
2. The design of fine-tuning from the image diffusion model is reasonable and effective.
3. The authors provide various experiments, including different conditions (text, image normal, and edge).
4. The presentation is clear and easy to follow.

**Weaknesses:**

1. The overall idea is not very interesting. DiffGS is another latent diffusion by changing the image vae to a 3D Gaussian vae. Only substituting the reconstruction module in LGM to a diffusion model lacks novelty. In fact, the 3D native diffusion model [1] has proved that we can generate very high-quality 3D objects by jointly representing the 3D shapes with geometry latents and PBR materials while the generated 3D Gaussians in DiffGS can only provide good renderings. The advantage of Gaussian Splatting representation is that it may contain higher-resolution information compared with implicit tokens in previous work. I am wondering if the authors could prove that the Gaussian-based latent diffusion can achieve more detailed generation. Besides, the poor geometry of 3D Gaussians is also a problem for applying this representation to 3D generation given that the 3D native diffusion has achieved such great progress.
2. It would be better if the authors could provide videos of the generated objects. It is hard to evaluate some 3D artifacts just by the current metrics (e.g., PSNR). For example, the Gaussians often show good rendering results on some views while suffering from obvious 3D inconsistency.  Some noisy Gaussians and artifacts can be easily observed in videos.
3. Applying a rendering loss combined with the diffusion loss is a common trick.

[1] CLAY: A Controllable Large-scale Generative Model for Creating High-quality 3D Assets

**Questions:**

I think some video demos are recommended for a 3D generation paper. Video demos are important to visually understand the 3D consistency and some geometric artifacts.

---

> ### Author Response · Authors · 2024-11-21
> **Response to Reviewer Xhe6 - Part 1**
>
> We sincerely appreciate the reviewer's insightful and valuable feedback. It is inspiring to know that you find our work efficient, with improved rendering quality, reasonable and effective method design, sufficient experiments, and an easy-to-follow presentation. Below are our clarifications regarding your concerns. We value your input and would welcome any follow-up questions or suggestions you may have. If you find our responses helpful, we sincerely hope the reviewer could kindly consider raising the score accordingly.
>
> **1.1. The proposed method is not very interesting. 3D native methods like CLAY show greater promise.**
>
> Thank you for bringing the impressive work CLAY [1] into the discussion. By leveraging proprietary internal 3D datasets and scaling up computational costs to train from scratch, CLAY has achieved remarkable performance in 3D object generation. However, we claim that this approach, which heavily relies on scaling up datasets manually created by digital artists, is unsustainable. Despite months of updates and the use of advanced training techniques such as RLHF, as discussed in **1.2** (the next question), limitations persist. This stems from the inherent challenges of creating high-quality man-made 3D assets, which are resource-intensive and lack the natural abundance seen in texts, images (videos), and audio.
>
> The core belief driving our work is that **multi-view images (videos) are the best 3D data source**. This is rooted in the strong connection between 3D content and images/videos, which are abundant, well-developed, and easily accessible. After all, humans live in a 3D world and possess the ability to perceive 3D structures without relying on explicit 3D models, instead interpreting the world through different viewpoints.
>
> The proposed method is inspired by the exciting progress of large image/video generative models trained on massive datasets, which have demonstrated the ability to perceive 3D concepts such as depth and normals [2,3,4,5]. Building on this, we take a step further by fine-tuning pretrained image generative models to directly generate 3D content. Each design choice in our approach emphasizes simplicity and explores the untapped potential of 2D generative priors, aiming to **harness the rapid advancements in image and video generation for 3D tasks**. We believe that combining the strengths of image/video priors is a promising and interesting direction for the future of 3D content generation.
>
> **1.2. Prove that the proposed Gaussian-based latent diffusion can achieve more detailed generation compared to 3D native methods like CLAY.**
>
> As noted by the reviewer, the 3DGS representation offers better renderings with higher-resolution details compared to 3D native models like CLAY, which separately represent 3D shapes using geometry and PBR materials. This limitation arises because CLAY relies on 3D meshes, which struggle to capture intricate and precise structures such as leaves, hair, and fur.
>
> To prove that, we evaluated a **commercial product** named [Rodin](https://hyperhuman.deemos.com/rodin) via membership subscription on its website, which is based on the CLAY technique, since CLAY itself is not open-sourced. The visualization results from these evaluations are provided in [https://iclr2025-paper1469.github.io/index.html#clay](https://iclr2025-paper1469.github.io/index.html#clay).
>
> It is worth noting that Rodin: (1) is trained on **proprietary internal datasets**, (2) originates from CLAY, which was initially trained on **256 A800 GPUs for 15 days**, (3) has undergone months of updates incorporating sophisticated techniques such as RLHF, (4) employs a complex generation pipeline composed of multiple distinct models (text-to-image, image-to-raw 3D, 3D object captioning and attribute prediction, 3D geometry refinement, PBR material generation and refinement, etc.), and (5) take about one minute to generate a 3D object though its pipeline. Despite these extensive resources and refinement steps, Rodin (Gen-1 RLHF V0.9) still **struggles to generate intricate and precise structures such as leaves and fur**, and **often fails to faithfully adhere to the provided image and text conditions**. In contrast, our method is **trained on open-source datasets using only 8 A100 GPUs over 2 to 5 days**, demonstrating greater efficiency and accessibility. Moreover, we believe incorporating refinement stages in the proposed method is a promising direction for future exploration.

---

> ### Author Response · Authors · 2024-11-21
> **Response to Reviewer Xhe6 - Part 2**
>
> **1.3. 3D Gaussians can only provide good renderings but with poor geometry, which is a problem for applications.**
>
> Thank you for bringing the applications of 3D generation to the discussion. We agree that 3D objects generated by CLAY exhibit better geometry than 3DGS for mesh representations, characterized by flat and neat surfaces, making them ideal for applications like 3D printing and animation. However, it is important to note that **creating individual 3D assets is not the only application for 3D generation**. Other applications, such as VR/AR immersive interaction, 3D game rendering, film production (particularly for far-away objects and backgrounds), and world simulation [11], **prioritize rendering quality and visual experience over geometric precision**. With the increasing adoption of simulation engines like Unity and Unreal Engine, which support real-time 3DGS rendering, we believe 3DGS holds significant potential for future applications. It should not be dismissed solely due to the great progress of 3D native diffusion that generates meshes.
>
> **2. Provide videos of the generated objects.**
>
> Thank you very much for the suggestion! We totally agree that video demos are important to visually understand the 3D consistency and some geometric artifacts. Videos of our generated 3D objects are provided in [https://iclr2025-paper1469.github.io](https://iclr2025-paper1469.github.io). We will release an official project page to provide video visualizations after the review period.
>
> **3. Applying a rendering loss with the diffusion loss is a common trick.**
>
> Thank you for bringing it up to the discussion. We agree that applying an additional rendering loss with the diffusion loss is indeed an idea that is easy to come up with. However, it is **not a common trick in 3D generative models** and is **only feasible with the other proposed design choices**.
>
> For example:
>
> - 3D native models, such as diffusion-based methods like CLAY [1], generate implicit 3D latents that are not renderable. Autoregressive-based methods like MeshGPT [6] produce 3D content sequentially without leveraging any 3D priors for rendering supervision.
>
> - Rendering-based models, including GAN-based GET3D [7] and diffusion-based DMV3D [8] can't access ground-truth 3D samples. As a result, they rely solely on rendering loss, which leads to unstable training and slower convergence.
>
> - Reconstruction-based models, such as LGM [9], generate multi-view images but are unable to perform 3D rendering. Methods like Ouroboros3D [10], which simultaneously optimize image generative models and 3D reconstruction models, require significantly more time and memory due to the need to propagate gradients through at least two large models, resulting in inefficient training.
>
> Incorporating rendering loss alongside the standard diffusion loss is a natural choice, as it complements the fine-tuning of an image generative model to process 3D latent representations efficiently renderable from arbitrary views. While we view this as **a byproduct of the proposed design choice rather than a core contribution**, the proposed models can still achieve competitive results without the rendering loss, as demonstrated by the ablation study on these two types of losses in **Table 5** of the original manuscript.
>
> Thank you again for mentioning ``rendering loss'' and we will add this discussion to the revised manuscript.
>
> ---
>
> [1] Zhang et al., CLAY: A Controllable Large-scale Generative Model for Creating High-quality 3D Assets, TOG 2024.
>
> [2] Ke et al., Repurposing Diffusion-Based Image Generators for Monocular Depth Estimation, CVPR 2024.
>
> [3] Fu et al., GeoWizard: Unleashing the Diffusion Priors for 3D Geometry Estimation from a Single Image, ECCV 2024.
>
> [4] Hu et al., DepthCrafter: Generating Consistent Long Depth Sequences for Open-world Videos, arXiv 2024.
>
> [5] Yang et al., Depth Any Video with Scalable Synthetic Data, arXiv 2024.
>
> [6] Siddiqui et al., MeshGPT: Generating Triangle Meshes with Decoder-Only Transformers, CVPR 2024.
>
> [7] Gao et al., GET3D: A Generative Model of High Quality 3D Textured Shapes Learned from Images, NeurIPS 2022.
>
> [8] Xu et al., DMV3D: Denoising Multi-View Diffusion using 3D Large Reconstruction Model, ICLR 2024.
>
> [9] Tang et al., LGM: Large Multi-View Gaussian Model for High-Resolution 3D Content Creation, ECCV 2024.
>
> [10] Wen et al., Ouroboros3D: Image-to-3D Generation via 3D-aware Recursive Diffusion, arXiv 2024.
>
> [11] Yu et al., Learning Visual Parkour from Generated Images, CORL 2024.

---

> > ### Comment · Reviewer_Xhe6 · 2024-11-26
> > **Response**
> >
> > Thanks for the rebuttal. The rebuttal provides additional comparisons with 3D-native methods and more detailed analyses. Most of my concerns are addressed. Therefore I will raise my score.

---

### Author Response · Authors · 2024-11-21
**General Response**

We would like to firmly express our gratitude to all reviewers for their insightful and constructive comments. It's greatly encouraging to note that **all reviewers agree** that (1) our contributions and presentation are good or excellent, (2) the proposed method is effective and efficient, and (3) experiments and ablation studies are thorough.

We respond to each reviewer with detailed individual replies. To address some of the concerns raised, we also include additional qualitative results in the anonymized link [https://iclr2025-paper1469.github.io](https://iclr2025-paper1469.github.io). We sincerely hope that our responses are helpful and informative. If you still have any concerns that we haven't addressed, we would love to hear from you. Please let us know what else we can do to further improve our work.

Thank you very much again for your time and input!

---

### Meta-Review · Area_Chair_KPVY · 2024-12-17

**Metareview:**

The paper introduces DiffGS, a framework for 3D Gaussian Splatting using image diffusion models. The reviewers acknowledged its efficient method design, comprehensive experiments, and good rendering quality. Key strengths include leveraging 2D priors effectively and maintaining 3D consistency. Weaknesses highlighted the lack of comparisons with amortized SDS methods and potential 3D inconsistencies in thin structures. Authors provided strong rebuttals, addressing most concerns and adding clarifications with new qualitative results. Given the overall positive reception and detailed responses, the AC recommends acceptance.

**Additional Comments On Reviewer Discussion:**

During rebuttal, reviewers raised concerns about comparisons with SDS-based methods, 3D consistency, and inference time. Authors addressed these with additional qualitative results, COLMAP-based evaluations, and inference time analysis. Concerns about thin-object artifacts and rendering losses were clarified with detailed explanations. Reviewers acknowledged the responses and raised scores. The paper's strengths in leveraging 2D priors for efficient 3D generation outweighed the limitations. These factors, alongside thorough author responses, justified the final decision to accept.

---

### Decision · Program_Chairs · 2025-01-22

Accept (Poster)